# Aleatoric and Epistemic Discrimination: Fundamental Limits of Fairness Interventions

**Hao Wang**
MIT-IBM Watson AI Lab
`hao@ibm.com`

**Luxi (Lucy) He**
Harvard College
`luxihe@college.harvard.edu`

**Rui Gao**
The University of Texas at Austin
`rui.gao@mccombs.utexas.edu`

**Flavio P. Calmon**
Harvard University
`flavio@seas.harvard.edu`

## Abstract

Machine learning (ML) models can underperform on certain population groups due to choices made during model development and bias inherent in the data. We categorize sources of discrimination in the ML pipeline into two classes: *aleatoric discrimination*, which is inherent in the data distribution, and *epistemic discrimination*, which is due to decisions made during model development. We quantify aleatoric discrimination by determining the performance limits of a model under fairness constraints, assuming perfect knowledge of the data distribution. We demonstrate how to characterize aleatoric discrimination by applying Blackwell's results on comparing statistical experiments. We then quantify epistemic discrimination as the gap between a model's accuracy when fairness constraints are applied and the limit posed by aleatoric discrimination. We apply this approach to benchmark existing fairness interventions and investigate fairness risks in data with missing values. Our results indicate that state-of-the-art fairness interventions are effective at removing epistemic discrimination on standard (overused) tabular datasets. However, when data has missing values, there is still significant room for improvement in handling aleatoric discrimination.

## 1 Introduction

Algorithmic discrimination may occur in different stages of the machine learning (ML) pipeline. For example, historical biases in the data-generating process can propagate to downstream tasks; human biases can influence a ML model through inductive bias; optimizing solely for accuracy can lead to disparate model performance across groups in the data (Suresh and Guttag, 2019; Mayson, 2019). The past years have seen a rapid increase in algorithmic interventions that aim to mitigate biases in ML models (see e.g., Zemel et al., 2013; Feldman et al., 2015; Calmon et al., 2017; Menon and Williamson, 2018; Zhang et al., 2018; Zafar et al., 2019; Friedler et al., 2019; Bellamy et al., 2019; Kim et al., 2019; Celis et al., 2019; Yang et al., 2020; Jiang and Nachum, 2020; Jiang et al., 2020; Martinez et al., 2020; Lowy et al., 2021; Alghamdi et al., 2022). A recent survey (Hort et al., 2022) found *nearly 400* fairness-intervention algorithms, including 123 pre-processing, 212 in-processing, and 56 post-processing algorithms introduced in the past decade.

Which sources of biases are (the hundreds of) existing fairness interventions trying to control? In order to create effective strategies for reducing algorithmic discrimination, it is critical to disentangle where biases in model performance originate. For instance, if a certain population group has significantly more missing features in training data, then it is more beneficial to collect data than selecting a more complex model class or training strategy. Conversely, if the model class does not accurately represent

37th Conference on Neural Information Processing Systems (NeurIPS 2023).

the underlying distribution of a specific population group, then collecting more data for that group will not resolve performance disparities.

We divide algorithmic discrimination[1] into two categories: aleatoric and epistemic discrimination.[2] Aleatoric discrimination captures inherent biases in the data distribution that can lead to unfair decisions in downstream tasks. Epistemic discrimination, in turn, is due to algorithmic choices made during model development and lack of knowledge about the optimal "fair" predictive model.

In this paper, we provide methods for measuring aleatoric and epistemic discrimination in classification tasks for group fairness metrics. Since aleatoric discrimination only depends on properties of the data distribution and the fairness measure of choice, we quantify it by asking a fundamental question:

*For a given data distribution, what is the best achievable performance (e.g., accuracy) under a set of group fairness constraints?*

We refer to the answer as the *fairness Pareto frontier*. This frontier delineates the optimal performance achievable by a classifier when unlimited data and computing power are available. For a fixed data distribution, the fairness Pareto frontier represents the ultimate, information-theoretic limit for accuracy and group fairness beyond which no model can achieve. Characterizing this limit enables us to (i) separate sources of discrimination and create strategies to control them accordingly; (ii) evaluate the effectiveness of existing fairness interventions for reducing epistemic discrimination; and (iii) inform the development of data collection methods that promote fairness in downstream tasks.

At first, computing the fairness Pareto frontier can appear to be an intractable problem since it requires searching over all possible classifiers—even if the data distribution is known exactly. Our main technical contribution is to provide an upper bound estimate for this frontier by solving a sequence of optimization problems. The proof technique is based on Blackwell's seminal results (Blackwell, 1953), which proposed the notion of comparisons of statistical experiments and inspired a line of works introducing alternative comparison criteria (see e.g., Shannon, 1958; Cam, 1964; Torgersen, 1991; Cohen et al., 1998; Raginsky, 2011). Here, we apply these results to develop an algorithm that iteratively refines the achievable fairness Pareto frontier. We also prove convergence guarantees for our algorithm and demonstrate how it can be used to benchmark existing fairness interventions.

We quantify epistemic discrimination by comparing a classifier's performance with the information-theoretic optimal given by the fairness Pareto frontier. Our experiments indicate that given sufficient data, state-of-the-art (SOTA) group fairness interventions are effective at reducing epistemic discrimination as their gap to the information-theoretic limit is small (see Figure 1 and 2). Consequently, there are diminishing returns in benchmarking new fairness interventions on standard (overused) tabular datasets (e.g., UCI Adult and ProPublica COMPAS datasets). However, existing interventions *do not* eliminate aleatoric discrimination as this type of discrimination is not caused by choice of learning algorithm or model class, and is due to the data distribution. Factors such as data missing values can significantly contribute to aleatoric discrimination. We observe that when population groups have disparate missing patterns, aleatoric discrimination escalates, leading to a sharp decline in the effectiveness of fairness intervention algorithms (see Figure 3).

**Related Work**

There is significant work analyzing the tension between group fairness measures and model performance metrics (see e.g., Kleinberg et al., 2016; Chouldechova, 2017; Corbett-Davies et al., 2017; Chen et al., 2018; Wick et al., 2019; Dutta et al., 2020; Wang et al., 2021). For example, there is a growing body of work on omnipredictors (Gopalan et al., 2021; Hu et al., 2023; Globus-Harris et al., 2023) discussing how, and under which conditions, the fair Bayes optimal classifier can be derived using post-processing techniques from multicalibrated regressors. While previous studies (Hardt et al., 2016; Corbett-Davies et al., 2017; Menon and Williamson, 2018; Chzhen et al., 2019; Yang et al., 2020; Zeng et al., 2022a,b) have investigated the fairness Pareto frontier and fair Bayes

---

[1]There are various measures to quantify algorithmic discrimination, and the choice should be based on the specific application of interest (see Blodgett et al., 2020; Varshney, 2021; Katzman et al., 2023, for a more detailed discussion). In this paper, we focus on group fairness measures (see Table 1 for some examples), which are crucial in contexts like hiring and recidivism prediction.

[2]We borrow this notion from ML uncertainty literature (see Hüllermeier and Waegeman, 2021, for a survey) and defer a detailed comparison in Appendix D.2.

optimal classifier, our approach differs from this prior work in the following aspects: our approach is applicable to *multiclass* classification problems with *multiple* protected groups; it *avoids disparate treatment* by not requiring the classifier to use group attributes as an input variable; and it can handle *multiple* fairness constraints simultaneously and produce fairness-accuracy trade-off curves (instead of a single point). Additionally, our proof techniques based on Blackwell's results on comparing statistical experiments are unique and may be of particular interest to fair ML and information theory communities. We present a detailed comparison with this line of work in Table 2 of Appendix E.

We recast the fairness Pareto frontier in terms of the conditional distribution $P_{\hat{Y}|Y,S}$ of predicted outcome $\hat{Y}$ given true label Y and group attributes S. This conditional distribution is related to confusion matrices conditioned on each subgroup. In this regard, our work is related to Verma and Rubin (2018); Alghamdi et al. (2020); Kim et al. (2020); Yang et al. (2020); Berk et al. (2021), which observed that many group fairness metrics can be written in terms of the confusion matrices for each subgroup. Among them, the closest work to ours is Kim et al. (2020), which optimized accuracy and fairness objectives over these confusion matrices and proposed a post-processing technique for training fair classifiers. However, they only imposed marginal sum constraints for the confusion matrices. We demonstrate that the feasible region of confusion matrices can be much smaller (see Remark 2 for an example), leading to a tighter approximation of the fairness Pareto frontier.

Recently, many strategies have been proposed to reduce the tension between group fairness and model performance by investigating properties of the data distribution. For example, Blum and Stangl (2019); Suresh and Guttag (2019); Fogliato et al. (2020); Wang et al. (2020); Mehrotra and Celis (2021); Fernando et al. (2021); Wang and Singh (2021); Zhang and Long (2021); Tomasev et al. (2021); Jacobs and Wallach (2021); Kallus et al. (2022); Jeong et al. (2022) studied how noisy or missing data affect fairness and model accuracy. Dwork et al. (2018); Ustun et al. (2019); Wang et al. (2021) considered training a separate classifier for each subgroup when their data distributions are different. Another line of research introduces data pre-processing techniques that manipulate data distribution for reducing its bias (e.g., Calmon et al., 2017; Kamiran and Calders, 2012). Among all these works, the closest one to ours is Chen et al. (2018), which decomposed group fairness measures into bias, variance, and noise (see their Theorem 1) and proposed strategies for reducing each term. Compared with Chen et al. (2018), the main difference is that we characterize a fairness Pareto frontier that depends on fairness metrics *and* a performance measure, giving a complete picture of how the data distribution influences fairness and accuracy.

## 2 Preliminaries

Next, we introduce notation, overview the key results in Blackwell (1953) on comparisons of experiments, and outline the fair classification setup considered in this paper.

**Notation.** For a positive integer $n$, let $[n] \triangleq \{1, \cdots, n\}$. We denote all probability distributions on the set $\mathcal{X}$ by $\mathcal{P}(\mathcal{X})$. Moreover, we define the probability simplex $\Delta_m \triangleq \mathcal{P}([m])$. When random variables A, X, Z form a Markov chain, we write $A - X - Z$. We write the mutual information between A, X as $I(A; X) \triangleq \mathbb{E}_{P_{A,X}} \left[ \log \frac{P_{A,X}(A,X)}{P_A(A)P_X(X)} \right]$. Since $I(A; X)$ is determined by the marginal distribution $P_A$ and the conditional distribution $P_{X|A}$, we also write $I(A; X)$ as $I(P_A; P_{X|A})$. When A, X are independent, we write $A \perp\!\!\!\perp X$.

If a random variable $A \in [n]$ has finite support, the conditional distribution $P_{X|A} : [n] \to \mathcal{P}(\mathcal{X})$ can be equivalently written as $\boldsymbol{P} \triangleq (P_1, \cdots, P_n)$ where each $P_i = P_{X|A=i} \in \mathcal{P}(\mathcal{X})$. Additionally, if $\mathcal{X}$ is a finite set $[m]$, then $P_{X|A}$ can be fully characterized by a transition matrix. We use $\mathcal{T}(m|n)$ to denote all transition matrices from $[n]$ to $[m]$: $\left\{ \boldsymbol{P} \in \mathbb{R}^{n \times m} \mid 0 \leq P_{i,j} \leq 1, \sum_{j=1}^m P_{i,j} = 1, \forall i \in [n] \right\}$.

### Comparisons of Experiments

Given two statistical experiments (i.e., conditional distributions) $\boldsymbol{P}$ and $\boldsymbol{Q}$, is there a way to decide which one is more informative? Here $\boldsymbol{P}$ and $\boldsymbol{Q}$ have the common input alphabet $[n]$ and potentially different output spaces. Blackwell gave an answer in his seminal work (Blackwell, 1953) from a decision-theoretic perspective. We review these results next.

| Fairness Metric | Abbr. | Definition
Expression w.r.t. $\boldsymbol{P}$ |
|---|---|---|
| Statistical Parity | SP $\leq \alpha_{\text{SP}}$ | $\|\Pr(\hat{Y} = \hat{y}\|S = s) - \Pr(\hat{Y} = \hat{y}\|S = s')\| \leq \alpha_{\text{SP}}$
$\left\|\sum_{y=1}^{C} \left(\frac{\mu_{s,y}}{\mu_s} P_{(s,y),\hat{y}} - \frac{\mu_{s',y}}{\mu_{s'}} P_{(s',y),\hat{y}}\right)\right\| \leq \alpha_{\text{SP}}$ |
| Equalized Odds | EO $\leq \alpha_{\text{EO}}$ | $\|\Pr(\hat{Y} = \hat{y}\|S = s, Y = y) - \Pr(\hat{Y} = \hat{y}\|S = s', Y = y)\| \leq \alpha_{\text{EO}}$
$\left\|P_{(s,y),\hat{y}} - P_{(s',y),\hat{y}}\right\| \leq \alpha_{\text{EO}}$ |
| Overall Accuracy Equality | OAE $\leq \alpha_{\text{OAE}}$ | $\|\Pr(\hat{Y} = Y\|S = s) - \Pr(\hat{Y} = Y\|S = s')\| \leq \alpha_{\text{OAE}}$
$\left\|\sum_{y=1}^{C} \left(\frac{\mu_{s,y}}{\mu_s} P_{(s,y),y} - \frac{\mu_{s',y}}{\mu_{s'}} P_{(s',y),y}\right)\right\| \leq \alpha_{\text{OAE}}$ |

Table 1: Standard group fairness metrics under multi-group and multi-class classification tasks. Here $\alpha_{\text{SP}}, \alpha_{\text{EO}}, \alpha_{\text{OAE}}, \in [0,1]$ are threshold parameters, $\hat{y}, y \in [C]$, $s, s' \in [A]$, and $\mu_{s,y}, \mu_s$ are defined in Proposition 1. Our analysis can be extended to many other group fairness metrics (see e.g., Table 1 in Kim et al., 2020).

Let $\mathcal{A}$ be a closed, bounded, convex subset of $\mathbb{R}^n$. A decision function $\boldsymbol{f}(x) = (a_1(x), \cdots, a_n(x))$ is any mapping from $\mathcal{X}$ to $\mathcal{A}$. It is associated with a loss vector:

$$\boldsymbol{v}(\boldsymbol{f}) = \left(\int a_1(x)\mathrm{d}P_1(x), \cdots, \int a_n(x)\mathrm{d}P_n(x)\right). \tag{1}$$

The collection of all $\boldsymbol{v}(\boldsymbol{f})$ is denoted by $\mathcal{B}(\boldsymbol{P}, \mathcal{A})$. Blackwell defined that $\boldsymbol{P}$ is more informative than $\boldsymbol{Q}$ if for every $\mathcal{A}$, $\mathcal{B}(\boldsymbol{P}, \mathcal{A}) \supseteq \mathcal{B}(\boldsymbol{Q}, \mathcal{A})$. Intuitively, this result means any risk achievable with $\boldsymbol{Q}$ is also achievable with $\boldsymbol{P}$. Moreover, Blackwell considered the standard measure $P^*$ which is the probability distribution of $\boldsymbol{p}(\bar{X})$ where $\boldsymbol{p}(x) : \mathcal{X} \to \Delta_n$ is a function defined as

$$\left(\frac{\mathrm{d}P_1}{\mathrm{d}P_1 + \cdots + \mathrm{d}P_n}, \cdots, \frac{\mathrm{d}P_n}{\mathrm{d}P_1 + \cdots + \mathrm{d}P_n}\right). \tag{2}$$

and $\bar{X}$ follows the probability distribution $\frac{P_1 + \cdots + P_n}{n}$. One of the most important findings by Blackwell in his paper is to discover the following equivalent conditions.

**Lemma 1** (Blackwell (1951, 1953))**.** *The following three conditions are equivalent:*

- *$\boldsymbol{P}$ is more informative than $\boldsymbol{Q}$;*

- *for any continuous and convex function $\phi : \Delta_n \to \mathbb{R}$, $\int \phi(\boldsymbol{p})dP^*(\boldsymbol{p}) \geq \int \phi(\boldsymbol{p})dQ^*(\boldsymbol{p})$;*

- *there is a stochastic transformation $\mathsf{T}$ s.t. $\mathsf{T}P_i = Q_i$. In other words, there exists a Markov chain $\mathrm{A} - \mathrm{X} - \mathrm{Z}$ for any distributions on $\mathrm{A}$ such that $\boldsymbol{P} = P_{\mathrm{X}|\mathrm{A}}$ and $\boldsymbol{Q} = P_{\mathrm{Z}|\mathrm{A}}$.*

If $\boldsymbol{P} = P_{\mathrm{X}|\mathrm{A}}$ is more informative than $\boldsymbol{Q} = P_{\mathrm{Z}|\mathrm{A}}$, by the third condition of Lemma 1 and the data processing inequality, $I(P_{\mathrm{A}}; P_{\mathrm{X}|\mathrm{A}}) \geq I(P_{\mathrm{A}}; P_{\mathrm{Z}|\mathrm{A}})$ holds for *any* marginal distribution $P_{\mathrm{A}}$. However, the converse does not hold in general—even if the above inequality holds for any $P_{\mathrm{A}}$, $\boldsymbol{P}$ is not necessarily more informative than $\boldsymbol{Q}$ (Rauh et al., 2017). In this regard, Blackwell's conditions are "stronger" than the mutual information based data processing inequality.

**Group Fair Classification**

Consider a multi-class classification task, where the goal is to train a probabilistic classifier $h : \mathcal{X} \to \Delta_C$ that uses input features X to predict their true label $Y \in [C]$. Additionally, assume the classifier produces a predicted outcome $\hat{Y} \in [C]$ and let $S \in [A]$ represent group attributes (e.g., race and sex). Depending on the domain of interest, X can either include or exclude S as an input to the classifier. Our framework can be easily extended to the setting where multiple subgroups overlap (Kearns et al., 2018). Throughout this paper, we focus on three standard group fairness measures: statistical parity (SP) (Feldman et al., 2015), equalized odds (EO) (Hardt et al., 2016; Pleiss et al., 2017), and overall accuracy equality (OAE) (Berk et al., 2021) (see Table 1 for their definitions) but our analysis can be extended to many other group fairness metrics, including the ones in Table 1 of Kim et al. (2020), as well as alternative performance measures beyond accuracy.

## 3 Fairness Pareto Frontier

In this section, we introduce our main concept—fairness Pareto frontier (FairFront). We use it to measure aleatoric discrimination and quantify epistemic discrimination by comparing a classifier's performance to the FairFront. We recast FairFront in terms of the conditional distribution $P_{\hat{Y}|S,Y}$ and apply Blackwell's conditions to characterize the feasible region of this conditional distribution. This effort converts a functional optimization problem into a convex program with a small number of variables. However, this convex program may involve infinitely many constraints. Hence, we introduce a greedy improvement algorithm that iteratively refines the approximation of FairFront and tightens the feasible region of $P_{\hat{Y}|S,Y}$. Finally, we establish a convergence guarantee for our algorithm.

Recall that we refer to aleatoric discrimination as the inherent biases of the data distribution that can lead to an unfair or inaccurate classifier. As its definition suggests, aleatoric discrimination only relies on properties of the data distribution and fairness metric of choice—it does not depend on the hypothesis class nor optimization method. Below we introduce FairFront that delineates a curve of optimal accuracy over all probabilistic classifiers under certain fairness constraints for a given data distribution $P_{S,X,Y}$. We use FairFront to quantify aleatoric discrimination.

**Definition 1.** For $\alpha_{\mathrm{SP}}, \alpha_{\mathrm{EO}}, \alpha_{\mathrm{OAE}} \geq 0$ and a given $P_{S,X,Y}$, we define $\mathsf{FairFront}(\alpha_{\mathrm{SP}}, \alpha_{\mathrm{EO}}, \alpha_{\mathrm{OAE}})$ by

$$\mathsf{FairFront}(\alpha_{\mathrm{SP}}, \alpha_{\mathrm{EO}}, \alpha_{\mathrm{OAE}}) \triangleq \max_{h} \ \mathbb{E}\left[\mathbb{I}_{\hat{Y}=Y}\right] \tag{3a}$$

$$\text{s.t. } \mathsf{SP} \leq \alpha_{\mathrm{SP}}, \mathsf{EO} \leq \alpha_{\mathrm{EO}}, \mathsf{OAE} \leq \alpha_{\mathrm{OAE}} \tag{3b}$$

where $\mathbb{I}$ is the indicator function; $\hat{Y}$ is produced by applying the classifier $h$ to X; the maximum is taken over all measurable $h$; and the definitions of SP, EO, and OAE are in Table 1. As a special case, if $\alpha_{\mathrm{SP}}, \alpha_{\mathrm{EO}}, \alpha_{\mathrm{OAE}} \geq 1$, then $\mathsf{FairFront}(\alpha_{\mathrm{SP}}, \alpha_{\mathrm{EO}}, \alpha_{\mathrm{OAE}})$ is the accuracy of the Bayes optimal classifier.

Solving this functional optimization problem is difficult since it optimizes over all measurable classifiers. There is a line of works that proposed different fairness-intervention algorithms for training group-fair classifiers (see e.g., Menon and Williamson, 2018; Zhang et al., 2018; Zafar et al., 2019; Celis et al., 2019; Yang et al., 2020; Wei et al., 2021; Alghamdi et al., 2022). They restrict the model class and vary loss functions and optimizers to find classifiers that approach FairFront as close as possible. However, these algorithms only describe a lower bound for FairFront. They do not determine what is the *best* achievable accuracy for a given set of fairness constraints.

We circumvent the above-mentioned challenges by rewriting FairFront in terms of the conditional distribution $P_{\hat{Y}|S,Y}$. The caveat is that although each classifier yields a $P_{\hat{Y}|S,Y}$, not every conditional distribution corresponds to a valid classifier. Hence, we introduce the following definition which characterizes all feasible $P_{\hat{Y}|S,Y}$.

**Definition 2.** Given $P_{X|S,Y}$, we define $\mathcal{C}$ as the set of all conditional distributions $P_{\hat{Y}|S,Y}$ where $\hat{Y}$ is produced by some probabilistic classifier $h$. In other words,

$$\mathcal{C} \triangleq \{P_{\hat{Y}|S,Y} \mid (S, Y) - X - \hat{Y}\}. \tag{4}$$

Throughout this paper, we write $P_{\hat{Y}|S,Y}$ or its corresponding transition matrix $\boldsymbol{P} \in \mathcal{T}(C|AC)$ interchangeably. Specifically, the $(C(s-1) + y)$-th row, $\hat{y}$-th column of $\boldsymbol{P}$ represents $P_{\hat{Y}|S,Y}(\hat{y}|s, y)$ and we denote it by $P_{(s,y),\hat{y}}$.

**Remark 1.** We demonstrate the connection between the conditional distribution $P_{\hat{Y}|S,Y}$ and confusion matrices in the setting of binary classification with binary groups. We define $\hat{\mathcal{C}}$ as the counterpart of $\mathcal{C}$ when we replace $P_{X|S,Y}$ with an empirical distribution $\hat{P}_{X|S,Y}$ computed from a dataset. The confusion matrix for group $s \in \{0, 1\}$ consists of four numbers: True Positive ($\mathsf{TP}_s$), False Positive ($\mathsf{FP}_s$), False Negative ($\mathsf{FN}_s$), True Negative ($\mathsf{TN}_s$). Assume that the number of positive-label data $n_s^+ = \mathsf{TP}_s + \mathsf{FN}_s$ and negative-label data $n_s^- = \mathsf{TN}_s + \mathsf{FP}_s$ are given—these numbers do not depend on the classifier. Then there is a one-to-one mapping from each element in $\hat{\mathcal{C}}$ to a confusion matrix:

$$\hat{P}_{\hat{Y}|S,Y}(1|s, 1) = \frac{1}{n_s^+}\mathsf{TP}_s, \quad \hat{P}_{\hat{Y}|S,Y}(1|s, 0) = \frac{1}{n_s^-}\mathsf{FP}_s,$$

$$\hat{P}_{\hat{Y}|S,Y}(0|s, 1) = \frac{1}{n_s^+}\mathsf{FN}_s, \quad \hat{P}_{\hat{Y}|S,Y}(0|s, 0) = \frac{1}{n_s^-}\mathsf{TN}_s.$$

Hence, $\hat{\mathcal{C}}$ essentially characterizes all feasible confusion matrices and $\mathcal{C}$ is the population counterpart of $\hat{\mathcal{C}}$. Note that $\mathcal{C}$ is determined by the underlying data distribution while $\hat{\mathcal{C}}$ (and confusion matrices) are tailored to a specific dataset.

**Proposition 1.** FairFront$(\alpha_{SP}, \alpha_{EO}, \alpha_{OAE})$ *in* (3) *is equal to the solution of the following convex optimization:*

$$\max_{\boldsymbol{P} \in \mathbb{R}^{AC \times C}} \sum_{s=1}^{A} \sum_{y=1}^{C} \mu_{s,y} P_{(s,y),y} \tag{5a}$$

$$\text{s.t. SP} \leq \alpha_{SP}, \text{EO} \leq \alpha_{EO}, \text{OAE} \leq \alpha_{OAE} \tag{5b}$$

$$\boldsymbol{P} \in \mathcal{C}. \tag{5c}$$

*Here the constants $\mu_{s,y} \triangleq \Pr(S = s, Y = y)$ and $\mu_s \triangleq \Pr(S = s)$ for $s \in [A]$, $y \in [A]$ and $P_{(s,y),\hat{y}}$ denotes the $(C(s - 1) + y)$-th row, $\hat{y}$-th column of the transition matrix $\boldsymbol{P}$, which is $P_{\hat{Y}|S,Y}(\hat{y}|s, y)$.*

For example, in binary classification with a binary group attribute, the above optimization only has 8 variables, 14 linear constraints + a single convex constraint $\boldsymbol{P} \in \mathcal{C}$. Hence, standard convex optimization solvers can directly compute its optimal value as long as we know how to characterize $\mathcal{C}$.

**Remark 2.** Note that Kim et al. (2020) investigated fairness Pareto frontiers via confusion matrices. The main difference is that Definition 1 in Kim et al. (2020) relaxed the constraint (5c) to $\boldsymbol{P} \in \mathcal{T}(C|AC)$ where $\mathcal{T}(C|AC)$ represents *all* transition matrices from $[AC]$ to $[C]$. This leads to a loose approximation of the frontier because $\mathcal{C}$ is often a strict subset of $\mathcal{T}(C|AC)$. To demonstrate this point, consider the scenario where $X \perp\!\!\!\perp (S, Y)$. Then $\hat{Y} \perp\!\!\!\perp (S, Y)$ by data processing inequality so

$$\mathcal{C} = \{\boldsymbol{P} \in \mathcal{T}(C|AC) \mid \text{each row of } \boldsymbol{P} \text{ is the same}\}. \tag{6}$$

Optimizing over $\mathcal{C}$ rather than $\mathcal{T}(C|AC)$ can significantly tighten the fairness Pareto frontier.

Before diving into the analysis, we first introduce a function $\boldsymbol{g} : \mathcal{X} \to \Delta_{AC}$ defined as $\boldsymbol{g}(x) = \left(P_{S,Y|X}(1, 1|x), \cdots, P_{S,Y|X}(A, C|x)\right)$. To obtain this function in practice, a common strategy among various post-processing fairness interventions (see e.g., Menon and Williamson, 2018; Alghamdi et al., 2022) is to train a probabilistic classifier that uses input features $X$ to predict $(S, Y)$. The output probability generated by this classifier is then utilized as an approximation of the function $\boldsymbol{g}$.

The following theorem is the main theoretical result in this paper. It provides a precise characterization of the set $\mathcal{C}$ through a series of convex constraints.

**Theorem 1.** *The set $\mathcal{C}$ is the collection of all transition matrices $\boldsymbol{P} \in \mathcal{T}(C|AC)$ such that the following condition holds:*
*For any $k \in \mathcal{N}$ and any $\{\boldsymbol{a}_i \mid \boldsymbol{a}_i \in [-1, 1]^{AC}, i \in [k]\}$,*

$$\sum_{\hat{y}=1}^{C} \max_{i \in [k]} \left\{\boldsymbol{a}_i^T \boldsymbol{\Lambda}_\mu \boldsymbol{p}_{\hat{y}}\right\} \leq \mathbb{E}\left[\max_{i \in [k]}\{\boldsymbol{a}_i^T \boldsymbol{g}(X)\}\right], \tag{7}$$

*where $\boldsymbol{p}_{\hat{y}}$ is the $\hat{y}$-th column of $\boldsymbol{P}$ and $\boldsymbol{\Lambda}_\mu = \mathsf{diag}(\mu_{1,1}, \cdots, \mu_{A,C})$.*

Intuitively, (7) uses piecewise linear functions to approximate the boundary of the convex set $\mathcal{C}$ where $k$ represents the number of linear pieces. Unfortunately, replacing $\boldsymbol{P} \in \mathcal{C}$ with this series of constraints in (5) may result in an intractable problem since standard duality-based approaches will lead to infinitely many dual variables. To resolve this issue, we first fix $k$ and let $\mathcal{C}_k$ be the set of $\boldsymbol{P}$ such that (7) holds under this fixed $k$. Accordingly, we define FairFront$_k(\alpha_{SP}, \alpha_{EO}, \alpha_{OAE})$ as the optimal value of (5) when replacing $\mathcal{C}$ with $\mathcal{C}_k$. Since $\mathcal{C}_1 \supseteq \mathcal{C}_2 \supseteq \cdots \supseteq \mathcal{C}$, we have FairFront$_1(\alpha_{SP}, \alpha_{EO}, \alpha_{OAE}) \geq$ FairFront$_2(\alpha_{SP}, \alpha_{EO}, \alpha_{OAE}) \geq \cdots \geq$ FairFront$(\alpha_{SP}, \alpha_{EO}, \alpha_{OAE})$. However, computing FairFront$_k(\alpha_{SP}, \alpha_{EO}, \alpha_{OAE})$ still involves infinitely many constraints.

Next, we introduce a greedy improvement algorithm that consists of solving a sequence of tractable optimization problems for approximating FairFront$_k(\alpha_{SP}, \alpha_{EO}, \alpha_{OAE})$. We use $\mathcal{A}$ to collect the constraints of $\boldsymbol{P}$ and set $\mathcal{A} = \emptyset$ initially. At each iteration, our algorithm solves a convex program to find an optimal $\boldsymbol{P}$ that maximizes the accuracy while satisfying the desired group fairness constraints and the constraints in $\mathcal{A}$; then we verify if this $\boldsymbol{P}$ is within the set $\mathcal{C}_k$ by solving a DC (difference of

**Algorithm 1** Approximate the fairness Pareto frontier.

---

**Input:** $\mathcal{D} = \{(x_i, y_i, s_i)\}_{i=1}^N$, max number of iterations $T$; max pieces $k$; classifier $g(x)$; $\alpha_{\text{SP}}, \alpha_{\text{EO}}, \alpha_{\text{OAE}}$.
**Initialize:** set $\mathcal{A} = \emptyset$; $\mu_{s,y} = \frac{|\{i \mid s_i=s, y_i=y\}|}{N}$; $t = 1$.
**Repeat:**
    Solve a convex program:

$$\max_{\boldsymbol{P}} \ \sum_{s=1}^{A} \sum_{y=1}^{C} \mu_{s,y} P_{(s,y),y}$$

$$\text{s.t. } \boldsymbol{P} \in \mathcal{T}(C \mid AC), \text{SP} \leq \alpha_{\text{SP}}, \text{EO} \leq \alpha_{\text{EO}}, \text{OAE} \leq \alpha_{\text{OAE}}$$

$$\sum_{\hat{y}=1}^{C} \max_{i \in [k]} \left\{ \boldsymbol{a}_i^T \boldsymbol{\Lambda}_\mu \boldsymbol{p}_{\hat{y}} \right\} \leq \mathbb{E} \left[ \max_{i \in [k]} \{ \boldsymbol{a}_i^T \boldsymbol{g}(\mathrm{X}) \} \right] \quad \forall (\boldsymbol{a}_1, \cdots, \boldsymbol{a}_k) \in \mathcal{A}.$$

    Let $v^t$ and $\boldsymbol{P}^t$ be the optimal value and optimal solution.
    Solve a DC program:

$$\min_{\substack{\boldsymbol{a}_i \in [-1,1]^{AC} \\ i \in [k]}} \mathbb{E} \left[ \max_{i \in [k]} \{ \boldsymbol{a}_i^T \boldsymbol{g}(\mathrm{X}) \} \right] - \sum_{\hat{y}=1}^{C} \max_{i \in [k]} \left\{ \boldsymbol{a}_i^T \boldsymbol{\Lambda}_\mu \boldsymbol{p}_{\hat{y}}^t \right\}.$$

**If** the optimal value is $\geq 0$ or $t = T$,
    **stop**;
**otherwise**,
    add the optimal $(\boldsymbol{a}_1, \cdots, \boldsymbol{a}_k)$ to $\mathcal{A}$ and $t = t + 1$.
**return:** $v^t, \boldsymbol{P}^t, \mathcal{A}$.

---

convex) program (Shen et al., 2016; Horst and Thoai, 1999). If $\boldsymbol{P} \in \mathcal{C}_k$, then the algorithm stops. Otherwise, the algorithm will find the constraint that is mostly violated by $\boldsymbol{P}$ and add this constraint to $\mathcal{A}$. Specifically, we determine a piecewise linear function that divides the space into two distinct regions: one containing $\boldsymbol{P}$ and the other containing $\mathcal{C}_k$. By "mostly violated", we mean the function is constructed to maximize the distance between $\boldsymbol{P}$ and the boundary defined by the function. We describe our algorithm in Algorithm 1 and establish a convergence guarantee below.

**Theorem 2.** *Let $T = \infty$. If Algorithm 1 stops, its output $\boldsymbol{P}^t$ is an optimal solution of* $\mathsf{FairFront}_k(\alpha_{\text{SP}}, \alpha_{\text{EO}}, \alpha_{\text{OAE}})$. *Otherwise, any convergent sub-sequence of $\{\boldsymbol{P}^t\}_{t=1}^{\infty}$ converges to an optimal solution of* $\mathsf{FairFront}_k(\alpha_{\text{SP}}, \alpha_{\text{EO}}, \alpha_{\text{OAE}})$.

Note that the output $v^t$ from Algorithm 1 is always an *upper bound* for $\mathsf{FairFront}(\alpha_{\text{SP}}, \alpha_{\text{EO}}, \alpha_{\text{OAE}})$, assuming the estimation error is sufficiently small. The tightness of this upper bound is determined by $k$ (i.e., how well the piecewise linear functions approximate the boundary of $\mathcal{C}$), $T$ (i.e., the total number of iterations). On the other hand, running off-the-shelf in-processing and post-processing fairness interventions can only yield *lower bounds* for $\mathsf{FairFront}(\alpha_{\text{SP}}, \alpha_{\text{EO}}, \alpha_{\text{OAE}})$.

## 4 Numerical Experiments

In this section, we demonstrate the tightness of our upper bound approximation of $\mathsf{FairFront}$, apply it to benchmark existing group fairness interventions, and show how data biases, specifically missing values, impact their effectiveness. We find that given sufficient data, SOTA fairness interventions are successful at reducing epistemic discrimination as their gap to (our upper bound estimate of) $\mathsf{FairFront}$ is small. However, we also discover that when different population groups have varying missing data patterns, aleatoric discrimination increases, which diminishes the performance of fairness intervention algorithms. Our numerical experiments are semi-synthetic since we apply fairness interventions to train classifiers using the *entire* dataset and resample from it as the test set. This setup enables us to eliminate the estimation error associated with Algorithm 1 (see Appendix E for a discussion). We provide additional experimental results and details in Appendix C.

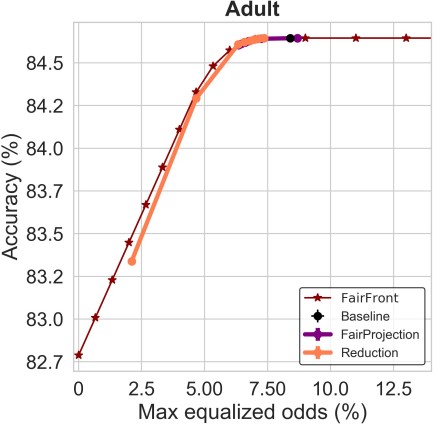
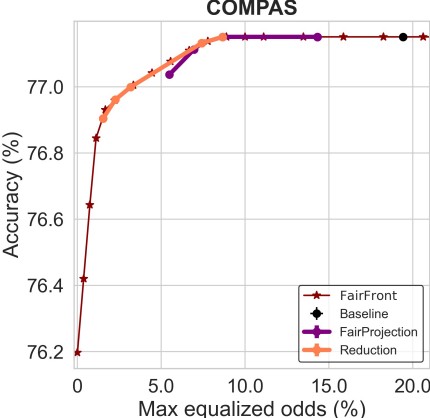

Figure 1: We compare `Reduction` and `FairProjection` with (our upper bound estimate of) FairFront on the Adult (Left) and COMPAS (Right) datasets. We train a classifier that approximates the Bayes optimal and use it as a basis for `Reduction` and `FairProjection`. This result not only demonstrates the tightness of our approximation but also shows that SOTA fairness interventions have already achieved near-optimal fairness-accuracy curves.

## 4.1 Benchmark Fairness Interventions

**Setup.** We evaluate our results on the UCI Adult dataset (Bache and Lichman, 2013), the ProPublica COMPAS dataset (Angwin et al., 2016), the German Credit dataset (Bache and Lichman, 2013), and HSLS (High School Longitudinal Study) dataset (Ingels et al., 2011; Jeong et al., 2022). We recognize that Adult, COMPAS, and German Credit datasets are overused and acknowledge the recent calls to move away from them (see e.g., Ding et al., 2021). We adopt these datasets for benchmarking purposes only since most fairness interventions have available code for these datasets. The HSLS dataset is a new dataset that first appeared in the fair ML literature last year and captures a common use-case of ML in education (student performance prediction, see Jeong et al., 2022). It has multi-class labels and multiple protected groups. We apply existing (group) fairness interventions to these datasets and measure their fairness violations via *Max equalized odds*:

$$\max \; |\Pr(\hat{Y} = \hat{y}|S = s, Y = y) - \Pr(\hat{Y} = \hat{y}|S = s', Y = y)|$$

where the max is taken over $y, \hat{y}, s, s'$. We run Algorithm 1 with $k = 6$ pieces, 20 iterations, and varying $\alpha_{\text{EO}}$ to estimate FairFront on each dataset. We compute the expectations and the $g$ function from the empirical distributions and solve the DC program by using the package in Shen et al. (2016). The details about how we pre-process these datasets and additional experimental results on the German Credit and HSLS datasets are deferred to Appendix C.

**Group fairness interventions.** We consider five existing fairness-intervention algorithms: `Reduction` (Agarwal et al., 2018), `EqOdds` (Hardt et al., 2016), `CalEqOdds` (Pleiss et al., 2017), `LevEqOpp` (Chzhen et al., 2019), and `FairProjection` Alghamdi et al. (2022). Among them, `Reduction` is an in-processing method and the rest are all post-processing methods. For the first three benchmarks, we use the implementations from IBM AIF360 library (Bellamy et al., 2018); for `LevEqOpp` and `FairProjection`, we use the Python implementations from the Github repo in Alghamdi et al. (2022). For `Reduction` and `FairProjection`, we can vary their tolerance of fairness violations to produce a fairness-accuracy curve; for `EqOdds`, `CalEqOdds`, and `LevEqOpp`, each of them produces a single point since they only allow hard equality constraint. We note that `FairProjection` is optimized for transforming probabilistic classifier outputs (see also Wei et al., 2021), but here we threshold the probabilistic outputs to generate binary predictions which may limit its performance. Finally, we train a random forest as the `Baseline` classifier.

**Results.** We observe that if we run Algorithm 1 for a single iteration, which is equivalent to solving Proposition 1 without (5c), its solution is very close to 1 for all $\alpha_{\text{EO}}$. This demonstrates the benefits of incorporating Blackwell's conditions into the fairness Pareto frontier.

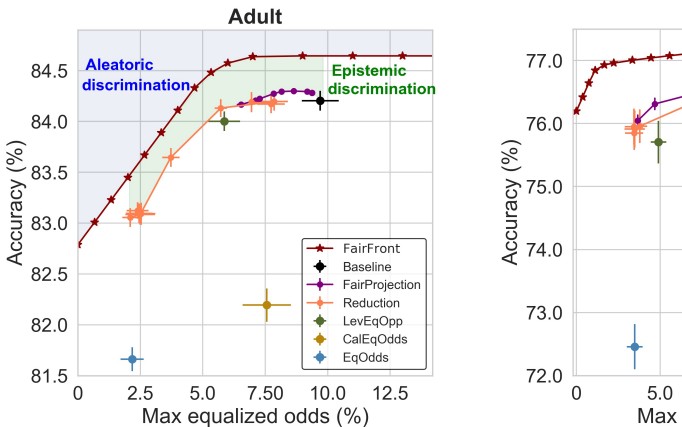

Figure 2: We benchmark existing fairness interventions using (our upper bound estimate of) FairFront. We use FairFront to quantify aleatoric discrimination and measure epistemic discrimination by comparing a classifier's accuracy and fairness violation with FairFront. The results show that SOTA fairness interventions are effective at reducing epistemic discrimination.

We train a classifier that approximates the Bayes optimal and use it as a basis for both `Reduction` and `FairProjection`, which are SOTA fairness interventions. We then apply these two fairness interventions to the entire dataset and evaluate their performance on the same dataset. Figure 1 shows that in this infinite sample regime, the fairness-accuracy curves produced by `Reduction` and `FairProjection` can approach our upper bound estimate of FairFront. This result not only demonstrates the tightness of our approximation (recall that Algorithm 1 gives an upper bound of FairFront and existing fairness interventions give lower bounds) but also shows that SOTA fairness interventions have already achieved near-optimal fairness-accuracy curves.

Recall that we use FairFront to quantify aleatoric discrimination since it characterizes the highest achievable accuracy among all classifiers satisfying the desired fairness constraints. Additionally, we measure epistemic discrimination by comparing a classifier's accuracy and fairness violation with FairFront. Given that our Algorithm 1 provides a tight approximation of FairFront, we use it to benchmark existing fairness interventions. Specifically, we first train a base classifier which may not achieve Bayes optimal accuracy. Then we use it as a basis for all existing fairness interventions. The results in Figure 2 show that SOTA fairness interventions remain effective at reducing epistemic discrimination. In what follows, we demonstrate how missing values in data can increase aleatoric discrimination and dramatically reduce the effectiveness of SOTA fairness interventions.

### 4.2 Fairness Risks in Missing Values

Real-world data often have missing values and the missing patterns can be different across different protected groups (see Jeong et al., 2022, for some examples). There is a growing line of research (see e.g., Jeong et al., 2022; Fernando et al., 2021; Wang and Singh, 2021; Subramonian et al., 2022; Caton et al., 2022; Zhang and Long, 2021; Schelter et al., 2019) studying the fairness risks of data with missing values. In this section, we apply our result to demonstrate how disparate missing patterns influence the fairness-accuracy curves.

**Setup.** We choose sex (group 0: female, group 1: male) as the group attribute for the Adult dataset, and race (group 0: African-American, group 1: Caucasian) for the COMPAS dataset. To investigate the impact of disparate missing patterns on aleatoric discrimination, we artificially generate missing values in both datasets. This is necessary as the datasets do not contain sufficient missing data. The missing values are generated according to different probabilities for different population groups. For each data point from group 0, we erase each input feature with a varying probability $p_0 \in \{10\%, 50\%, 70\%\}$, while for group 1, we erase each input feature with a fixed probability $p_1 = 10\%$. We then apply mode imputation to the missing values, replacing them with the mode of non-missing values for each feature. Finally, we apply Algorithm 1 along with `Reduction` and `Baseline` to the imputed data. The experimental results are shown in Figure 3.

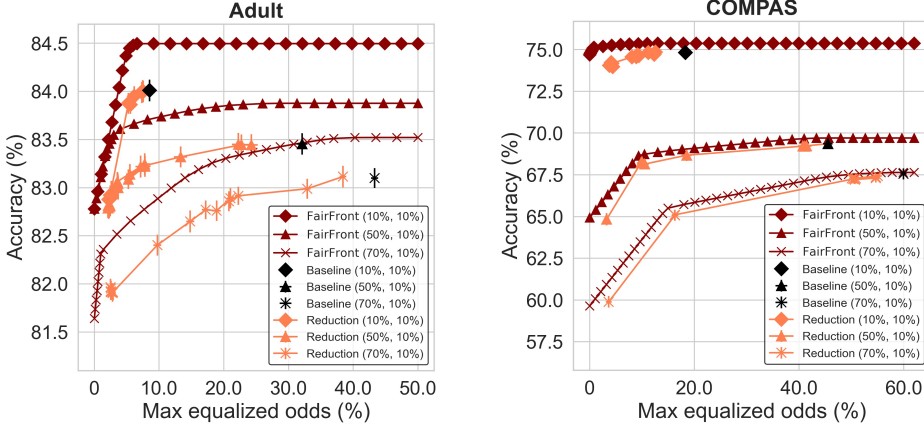

Figure 3: Fairness risks of disparate missing patterns. The missing probabilities of group 0 (female in Adult/African-American in COMPAS) and group 1 (male in Adult/Caucasian in COMPAS) are varying among $\{(10\%, 10\%), (50\%, 10\%), (70\%, 10\%)\}$. We apply `Reduction` and `Baseline` to the imputed data and plot their fairness-accuracy curves against `FairFront`. As shown, the effectiveness of fairness interventions substantially decrease with increasing disparate missing patterns in data.

**Results.**    As we increase the missing probability of group 0, (our upper bound estimate of) `FairFront` decreases since it becomes more difficult to accurately predict outcomes for group 0. This in turn affects the overall model performance, since the fairness constraint requires that the model performs similarly for both groups. We also observe the fairness-accuracy curves of `Reduction` decrease as the missing data for group 0 become more prevalent. In other words, as the missing data for group 0 increase, it becomes more difficult to maintain both high accuracy and fairness in the model's prediction.

## 5    Final Remarks

The past years have witnessed a growing line of research introducing various group fairness-intervention algorithms. Most of these interventions focus on optimizing model performance subject to group fairness constraints. Though comparing and benchmarking these methods on various datasets is valuable (e.g., see benchmarks in Friedler et al., 2019; Bellamy et al., 2019; Wei et al., 2021), this does not reveal if there is still room for improvement in their fairness-accuracy curves, or if existing methods approach the information-theoretic optimal limit when infinite data is available. Our results address this gap by introducing the fairness Pareto frontier, which measures the highest possible accuracy under a set of group fairness constraints. We precisely characterize the fairness Pareto frontier using Blackwell's conditions and present a greedy improvement algorithm that approximates it from data. Our results show that the fairness-accuracy curves produced by SOTA fairness interventions are very close to the fairness Pareto frontier on standard datasets.

Additionally, we demonstrate that when data are biased due to missing values, the fairness Pareto frontier degrades. Although existing fairness interventions can still reduce performance disparities, they come at the cost of significantly lowering overall model accuracy. The methods we present for computing the fairness Pareto frontier can also be applied to analyze other sources of aleatoric discrimination, such as when individuals may misreport their data or when there are measurement errors. Overall, the fairness Pareto frontier can serve as a valuable framework for guiding data collection and cleaning. Our results indicate that existing fairness interventions can be effective in reducing epistemic discrimination, and there are diminishing returns in developing new fairness interventions focused solely on optimizing accuracy for a given group fairness constraint on pristine data. However, existing fairness interventions have yet to effectively provide both fair and accurate classification when additional sources of aleatoric discrimination are present (such as missing values in data). This suggests that there is still significant need for research on handling aleatoric sources of discrimination that appear throughout the data collection process.

We provide an in-depth discussion on future work in Appendix E.

## Acknowledgement

This material is based upon work supported by the National Science Foundation under grants CAREER 1845852, CIF 2312667, FAI 2040880, CIF 1900750.

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

# A  Technical Background

In this section, we extend some results in Blackwell (1951, 1953) to our setting. For a random variable X, we denote its probability distribution by $\mathcal{L}(X)$. A conditional distribution $P_{X|A} : [n] \to \mathcal{P}(\mathcal{X})$ can be equivalently written as $\boldsymbol{P} \triangleq (P_1, \cdots, P_n)$ where each $P_i = P_{X|A=i} \in \mathcal{P}(\mathcal{X})$. Let $\mathcal{A}$ be a closed, bounded, convex subset of $\mathbb{R}^n$. A decision function is a mapping $\boldsymbol{f} : \mathcal{X} \to \mathcal{A}$, which can also be written as $\boldsymbol{f}(x) = (a_1(x), \cdots, a_n(x))$. A decision function is associated a loss vector:

$$\boldsymbol{v}(\boldsymbol{f}) = \left( \int a_1(x) \mathrm{d}P_1(x), \cdots, \int a_n(x) \mathrm{d}P_n(x) \right). \tag{9}$$

The collection of all $\boldsymbol{v}(\boldsymbol{f})$ is denoted by $\mathcal{B}(P_{X|A}, \mathcal{A})$ or $\mathcal{B}(\boldsymbol{P}, \mathcal{A})$.

For a vector $\boldsymbol{\lambda} \in \Delta_n$ such that $\boldsymbol{\lambda} > 0$, we define a function $\boldsymbol{p_\lambda}(x) : \mathcal{X} \to \Delta_n$:

$$\boldsymbol{p_\lambda}(x) = \left( \frac{\lambda_1 \mathrm{d}P_1}{\lambda_1 \mathrm{d}P_1 + \cdots + \lambda_n \mathrm{d}P_n}, \cdots, \frac{\lambda_n \mathrm{d}P_n}{\lambda_1 \mathrm{d}P_1 + \cdots + \lambda_n \mathrm{d}P_n} \right). \tag{10}$$

Note that $\boldsymbol{p_\lambda}(X)$ is a sufficient statistic for X, considering A as the parameter (it can be proved by Fisher-Neyman factorization theorem). In other words, two Markov chains hold: $A - \boldsymbol{p_\lambda}(X) - X$ and $A - X - \boldsymbol{p_\lambda}(X)$ for any distribution on A.

Consider a new set of probability distributions $\boldsymbol{P_\lambda^*} \triangleq (\mathcal{L}(\boldsymbol{p_\lambda}(X_1)), \cdots, \mathcal{L}(\boldsymbol{p_\lambda}(X_n)))$ where $\mathcal{L}(X_i) = P_i$. Here $\boldsymbol{P_\lambda^*}$ can be viewed as a conditional distribution from $[n]$ to $\mathcal{P}(\Delta_n)$ since each $\mathcal{L}(\boldsymbol{p_\lambda}(X_i))$ is a probability distribution over $\Delta_n$. The following lemma follows from the sufficiency of $\boldsymbol{p_\lambda}(X)$.

**Lemma 2** (Adaptation of Theorem 3 in Blackwell (1951)). *For any $\mathcal{A}$, $\mathcal{B}(\boldsymbol{P}, \mathcal{A}) = \mathcal{B}(\boldsymbol{P_\lambda^*}, \mathcal{A})$.*

*Proof.* Suppose that $\boldsymbol{f}^*(\boldsymbol{p}) = (a_1^*(\boldsymbol{p}), \cdots, a_n^*(\boldsymbol{p}))$ is a decision function for $(\boldsymbol{P_\lambda^*}, \mathcal{A})$. Accordingly, we define $\boldsymbol{f}(x) = (a_1^*(\boldsymbol{p_\lambda}(x)), \cdots, a_n(\boldsymbol{p_\lambda}(x)))$ where the function $\boldsymbol{p_\lambda}$ is defined in (10). Then it is clear that $\boldsymbol{f}$ is a decision function for $(\boldsymbol{P}, \mathcal{A})$. By the law of unconscious statistician, we have

$$\int a_i^*(\boldsymbol{p}) \mathrm{d}P_{\boldsymbol{\lambda}, i}^*(\boldsymbol{p}) = \mathbb{E}\left[ a_i^*(\boldsymbol{p_\lambda}(X_i)) \right] = \int a_i^*(\boldsymbol{p_\lambda}(x)) \mathrm{d}P_i(x). \tag{11}$$

Hence, $\boldsymbol{v}(\boldsymbol{f}^*) = \boldsymbol{v}(\boldsymbol{f})$, which implies $\mathcal{B}(\boldsymbol{P_\lambda^*}, \mathcal{A}) \subseteq \mathcal{B}(\boldsymbol{P}, \mathcal{A})$. For the other direction, suppose $\boldsymbol{f}(x) = (a_1(x), \cdots, a_n(x))$ is a decision function for $(\boldsymbol{P}, \mathcal{A})$. Let $\boldsymbol{f}^*(\boldsymbol{p}) = (a_1^*(\boldsymbol{p}), \cdots, a_n^*(\boldsymbol{p}))$ where $a_i^*(\boldsymbol{p}) \triangleq \mathbb{E}\left[ a_i(X_i) \mid \boldsymbol{p_\lambda}(X_i) = \boldsymbol{p} \right]$. Since $\boldsymbol{p_\lambda}(X)$ is a sufficient statistics, for any $i \in [n]$

$$\mathcal{L}(X_i | \boldsymbol{p_\lambda}(X_i) = \boldsymbol{p}) = \mathcal{L}(X_1 | \boldsymbol{p_\lambda}(X_1) = \boldsymbol{p}). \tag{12}$$

Therefore, $\boldsymbol{f}^*(\boldsymbol{p}) = \mathbb{E}\left[ \boldsymbol{f}(X_1) | \boldsymbol{p_\lambda}(X_1) = \boldsymbol{p} \right]$. Since $\mathcal{A}$ is a convex set, $\boldsymbol{f}^*$ is a decision function for $(\boldsymbol{P}^*, \mathcal{A})$. By the law of total expectation, we have

$$\int a_i^*(\boldsymbol{p}) \mathrm{d}P_{\boldsymbol{\lambda}, i}^*(\boldsymbol{p}) = \int a_i(x) \mathrm{d}P_i(x). \tag{13}$$

Hence, $\boldsymbol{v}(\boldsymbol{f}) = \boldsymbol{v}(\boldsymbol{f}^*)$, which implies $\mathcal{B}(\boldsymbol{P}, \mathcal{A}) \subseteq \mathcal{B}(\boldsymbol{P_\lambda^*}, \mathcal{A})$. $\square$

For a vector $\boldsymbol{\lambda} \in \Delta_n$ such that $\boldsymbol{\lambda} > 0$, the condition distribution $P_{X|A}$ induces a weighted standard measure $P_{\boldsymbol{\lambda}}^* \triangleq \mathcal{L}(\boldsymbol{p_\lambda}(\bar{X}))$ where $\mathcal{L}(\bar{X}) = \lambda_1 P_1 + \cdots + \lambda_n P_n$.

**Theorem 3** (Adaptation of Theorem 4 in Blackwell (1951)). *For any two conditional distributions $P_{X|A}$ and $Q_{Y|A}$, let $P_{\boldsymbol{\lambda}}^*$ and $Q_{\boldsymbol{\lambda}}^*$ be their weighted standard measures, respectively. Then $\mathcal{B}(P_{X|A}, \mathcal{A}) \supseteq \mathcal{B}(Q_{Y|A}, \mathcal{A})$ for any closed, bounded, convex set $\mathcal{A}$ if and only if for any continuous convex $\phi : \Delta_n \to \mathbb{R}$, $\int \phi(\boldsymbol{p}) dP_{\boldsymbol{\lambda}}^*(\boldsymbol{p}) \geq \int \phi(\boldsymbol{p}) dQ_{\boldsymbol{\lambda}}^*(\boldsymbol{p})$*

*Proof.* First, by Lemma 2, we know $\mathcal{B}(P_{X|A}, \mathcal{A}) = \mathcal{B}(\boldsymbol{P_\lambda^*}, \mathcal{A})$ and $\mathcal{B}(Q_{Y|A}, \mathcal{A}) = \mathcal{B}(\boldsymbol{Q_\lambda^*}, \mathcal{A})$. We denote $\boldsymbol{\Lambda} = \mathrm{diag}(\lambda_1, \cdots, \lambda_n)$. Consider any $\mathcal{A} = \mathrm{conv}(\boldsymbol{a}_1, \cdots, \boldsymbol{a}_k)$. Let

$$\boldsymbol{f}^*(\boldsymbol{p}) = \operatorname*{argmin}_{\boldsymbol{a} \in \mathcal{A}} \boldsymbol{p}^T \boldsymbol{\Lambda}^{-1} \boldsymbol{a}. \tag{14}$$

Note that $\boldsymbol{f}^*(\boldsymbol{p}) \in \{\boldsymbol{a}_1, \cdots, \boldsymbol{a}_k\}$ since this set contains all the extreme points of $\mathcal{A}$.[3] By definition, for any decision function w.r.t. $(\boldsymbol{P}_{\boldsymbol{\lambda}}^*, \mathcal{A})$, we have

$$\boldsymbol{p}^T \boldsymbol{\Lambda}^{-1} \boldsymbol{f}(\boldsymbol{p}) \geq \boldsymbol{p}^T \boldsymbol{\Lambda}^{-1} \boldsymbol{f}^*(\boldsymbol{p}), \quad \forall \boldsymbol{p}. \tag{15}$$

Let $\boldsymbol{v} = \boldsymbol{v}(\boldsymbol{f})$. By the same reason with (11), we have

$$v_j = \int a_j(\boldsymbol{p_\lambda}(x)) \mathrm{d}P_j(x) \tag{16}$$

$$= \frac{1}{\lambda_j} \int a_j(\boldsymbol{p_\lambda}(x)) \frac{\lambda_j \mathrm{d}P_j}{\lambda_1 \mathrm{d}P_1 + \cdots + \lambda_n \mathrm{d}P_n}(x)(\lambda_1 \mathrm{d}P_1 + \cdots + \lambda_n \mathrm{d}P_n)(x) \tag{17}$$

$$= \frac{1}{\lambda_j} \int a_j(\boldsymbol{p_\lambda}(x))[\boldsymbol{p_\lambda}(x)]_j (\lambda_1 \mathrm{d}P_1 + \cdots + \lambda_n \mathrm{d}P_n)(x) \tag{18}$$

$$= \frac{1}{\lambda_j} \mathbb{E}\left[a_j(\boldsymbol{p_\lambda}(\bar{\mathrm{X}}))[\boldsymbol{p_\lambda}(\bar{\mathrm{X}})]_j\right] \tag{19}$$

$$= \frac{1}{\lambda_j} \int a_j(\boldsymbol{p}) p_j \mathrm{d}P_{\boldsymbol{\lambda}}^*(\boldsymbol{p}), \tag{20}$$

where the last step is due to the law of unconscious statistician. Therefore,

$$\sum_{j=1}^n v_j = \int \boldsymbol{p}^T \boldsymbol{\Lambda}^{-1} \boldsymbol{f}(\boldsymbol{p}) \mathrm{d}P_{\boldsymbol{\lambda}}^*(\boldsymbol{p}) \tag{21}$$

$$\geq \int \boldsymbol{p}^T \boldsymbol{\Lambda}^{-1} \boldsymbol{f}^*(\boldsymbol{p}) \mathrm{d}P_{\boldsymbol{\lambda}}^*(\boldsymbol{p}) \tag{22}$$

$$= \int \min_i \{\boldsymbol{p}^T \boldsymbol{\Lambda}^{-1} \boldsymbol{a}_i\} \mathrm{d}P_{\boldsymbol{\lambda}}^*(\boldsymbol{p}). \tag{23}$$

The equality is achieved by $\boldsymbol{v}(\boldsymbol{f}^*)$. Hence, for any $\mathcal{A} = \mathrm{conv}(\boldsymbol{a}_1, \cdots, \boldsymbol{a}_k)$

$$\min_{\boldsymbol{v} \in \mathcal{B}(P_{\mathrm{X}|\mathrm{A}}, \mathcal{A})} \sum_{j=1}^n v_j = \int \min_i \{\boldsymbol{a}_i^T \boldsymbol{\Lambda}^{-1} \boldsymbol{p}\} \mathrm{d}P_{\boldsymbol{\lambda}}^*(\boldsymbol{p}). \tag{24}$$

Recall that Theorem 2.(3) in Blackwell (1951) states

$$\mathcal{B}(P_{\mathrm{X}|\mathrm{A}}, \mathcal{A}) \supseteq \mathcal{B}(P_{\mathrm{Y}|\mathrm{A}}, \mathcal{A}) \quad \text{for every closed, bounded, convex } \mathcal{A}$$

$$\Leftrightarrow \min_{\boldsymbol{v} \in \mathcal{B}(P_{\mathrm{X}|\mathrm{A}}, \mathcal{A})} \sum_{j=1}^n v_j \leq \min_{\boldsymbol{v} \in \mathcal{B}(P_{\mathrm{Y}|\mathrm{A}}, \mathcal{A})} \sum_{j=1}^n v_j \quad \text{for every closed, bounded, convex } \mathcal{A}.$$

By approximation theory, the second condition can be relaxed to any $\mathcal{A}$ that is a convex hull of a finite set. By (24), this relaxed condition is equivalent to

$$\int \phi(\boldsymbol{p}) \mathrm{d}P_{\boldsymbol{\lambda}}^*(\boldsymbol{p}) \geq \int \phi(\boldsymbol{p}) \mathrm{d}Q_{\boldsymbol{\lambda}}^*(\boldsymbol{p}) \tag{25}$$

for all $\phi(\boldsymbol{p})$ that are the **maximum** of finitely many linear functions. By approximation theory again, the above condition is equivalent to the one holding for any continuous convex function $\phi$. $\qquad \square$

## B  Omitted Proofs

### B.1  Proof of Lemma 3

*Proof.* Clearly, $\mathcal{C}$ is a subset of $\mathcal{T}(C|AC)$. Let $\lambda \in (0, 1)$ and $P_{\hat{\mathrm{Y}}_0|\mathrm{S},\mathrm{Y}}, P_{\hat{\mathrm{Y}}_1|\mathrm{S},\mathrm{Y}} \in \mathcal{C}$. Now we introduce a Bernoulli random variable B such that $\Pr(\mathrm{B} = 0) = \lambda$. Finally, we define $\hat{\mathrm{Y}}_\lambda = \mathrm{B}\hat{\mathrm{Y}}_1 + (1 - \mathrm{B})\hat{\mathrm{Y}}_0$. By definition, we have $(\mathrm{S}, \mathrm{Y}) - \mathrm{X} - \hat{\mathrm{Y}}_\lambda$ so $P_{\hat{\mathrm{Y}}_\lambda|\mathrm{S},\mathrm{Y}} \in \mathcal{C}$. Moreover,

$$P_{\hat{\mathrm{Y}}_\lambda|\mathrm{S},\mathrm{Y}} = \lambda P_{\hat{\mathrm{Y}}_0|\mathrm{S},\mathrm{Y}} + (1 - \lambda) P_{\hat{\mathrm{Y}}_1|\mathrm{S},\mathrm{Y}}.$$

---

[3]If (14) has multiple optimal solutions, we always choose the one from $\{\boldsymbol{a}_1, \cdots, \boldsymbol{a}_k\}$.

Hence, $\mathcal{C}$ is convex.

Let $\lambda \in (0,1)$. Assume $\boldsymbol{P}$ and $\bar{\boldsymbol{P}}$ achieve the maximal values of Proposition 1 under $(\alpha_{\text{SP}}, \alpha_{\text{EO}}, \alpha_{\text{OAE}})$ and $(\bar{\alpha}_{\text{SP}}, \bar{\alpha}_{\text{EO}}, \bar{\alpha}_{\text{OAE}})$, respectively. We define $\boldsymbol{P}_\lambda = \lambda \boldsymbol{P} + (1-\lambda)\bar{\boldsymbol{P}}$, which satisfies the constraints of Proposition 1 with thresholds $(\lambda \alpha_{\text{SP}} + (1-\lambda)\bar{\alpha}_{\text{SP}}, \lambda \alpha_{\text{EO}} + (1-\lambda)\bar{\alpha}_{\text{EO}}, \lambda \alpha_{\text{OAE}} + (1-\lambda)\bar{\alpha}_{\text{OAE}})$. Finally, since the objective function of Proposition 1 is a linear function, it is equal to $\lambda \mathsf{FairFront}(\alpha_{\text{SP}}, \alpha_{\text{EO}}, \alpha_{\text{OAE}}) + (1-\lambda)\mathsf{FairFront}(\bar{\alpha}_{\text{SP}}, \bar{\alpha}_{\text{EO}}, \bar{\alpha}_{\text{OAE}})$ under $\boldsymbol{P}_\lambda$. $\qquad\square$

### B.2  Proof of Theorem 1

*Proof.* The proof relies on Theorem 3 and Lemma 1. For simplicity, we write the conditional $P_{\hat{\text{Y}}|\text{S},\text{Y}}$ as its corresponding transition matrix $\boldsymbol{P}$. Let $\boldsymbol{\mu} = (\Pr(\text{S}=1, \text{Y}=1), \cdots, \Pr(\text{S}=A, \text{Y}=C))$. The function (10) in our setting can be written as:

$$\boldsymbol{p_\mu}(\hat{y}) = \left( \frac{\mu_{1,1} P_{(1,1),\hat{y}}}{\sum_{s,y} \mu_{s,y} P_{(s,y),\hat{y}}}, \cdots, \frac{\mu_{A,C} P_{(A,C),\hat{y}}}{\sum_{s,y} \mu_{s,y} P_{(s,y),\hat{y}}} \right). \tag{26}$$

$$\boldsymbol{p_\mu}(x) = \left( \frac{\mu_{1,1} \mathrm{d}P_{\text{X}|\text{S}=1,\text{Y}=1}}{\sum_{s,y} \mu_{s,y} \mathrm{d}P_{\text{X}|\text{S}=s,\text{Y}=y}}(x), \cdots, \frac{\mu_{A,C} \mathrm{d}P_{\text{X}|\text{S}=A,\text{Y}=C}}{\sum_{s,y} \mu_{s,y} \mathrm{d}P_{\text{X}|\text{S}=s,\text{Y}=y}}(x) \right). \tag{27}$$

Note that $\boldsymbol{p_\mu}(x) = \boldsymbol{g}(x)$ due to Bayes' rule. By Lemma 1, we can rewrite $\mathcal{C}$ in Definition 2 as

$$\mathcal{C} = \left\{ \boldsymbol{P} \mid P_{\hat{\text{X}}|\text{S},\text{Y}} \text{ is more informative than } \boldsymbol{P} \right\}. \tag{28}$$

By Lemma 1 and Theorem 3, the above set is further equivalent to all transition matrices $\boldsymbol{P} \in \mathcal{T}(C|AC)$ satisfying

$$\sum_{\hat{y}=1}^{C} \phi \left( \frac{\mu_{1,1} P_{(1,1),\hat{y}}}{\sum_{s,y} \mu_{s,y} P_{(s,y),\hat{y}}}, \cdots, \frac{\mu_{A,C} P_{(A,C),\hat{y}}}{\sum_{s,y} \mu_{s,y} P_{(s,y),\hat{y}}} \right) \sum_{s,y} \mu_{s,y} P_{(s,y),\hat{y}} \le \mathbb{E}\left[ \phi(\boldsymbol{g}(\text{X})) \right] \tag{29}$$

for any function $\phi: \Delta_{AC} \to \mathbb{R}$ which is the maximum of finitely many linear functions. Now we can write $\phi(\boldsymbol{p}) = \max_{i \in [k]} \left\{ \boldsymbol{a}_i^T \boldsymbol{p} \right\}$—we ignore the bias term because $\boldsymbol{a}_i^T \boldsymbol{p} + b_i = (\boldsymbol{a}_i + b_i \boldsymbol{1})^T \boldsymbol{p}$. Then the inequality in (29) can be simplified as

$$\sum_{\hat{y}=1}^{C} \max_{i \in [k]} \left\{ \boldsymbol{a}_i^T \boldsymbol{\Lambda}_\mu \boldsymbol{p}_{\hat{y}} \right\} \le \mathbb{E}\left[ \max_{i \in [k]} \{ \boldsymbol{a}_i^T \boldsymbol{g}(\text{X}) \} \right], \tag{30}$$

where $\boldsymbol{p}_{\hat{y}}$ is the $\hat{y}$-th column of $\boldsymbol{P}$ and $\boldsymbol{\Lambda}_\mu = \mathsf{diag}(\mu_{1,1}, \cdots, \mu_{A,C})$. Finally, we can always normalize the above inequality so that each $\boldsymbol{a}_i \in [-1,1]^{AC}$. $\qquad\square$

### B.3  Proof of Theorem 2

*Proof.* We denote

$$f(\boldsymbol{P}) \triangleq \sum_{s=1}^{A} \sum_{y=1}^{C} \mu_{s,y} P_{(s,y),y},$$

$$g(\boldsymbol{P}; \boldsymbol{a}_1, \cdots, \boldsymbol{a}_k) \triangleq \sum_{\hat{y}=1}^{C} \max_{i \in [k]} \left\{ \boldsymbol{a}_i^T \boldsymbol{\Lambda}_\mu \boldsymbol{p}_{\hat{y}} \right\} - \mathbb{E}\left[ \max_{i \in [k]} \{ \boldsymbol{a}_i^T \boldsymbol{g}(\text{X}) \} \right],$$

$$\mathcal{F} \triangleq \mathcal{C}_k \cap \{ \boldsymbol{P} \in \mathcal{T}(C|AC) \mid \mathsf{SP} \le \alpha_{\text{SP}}, \mathsf{EO} \le \alpha_{\text{EO}}, \mathsf{OAE} \le \alpha_{\text{OAE}} \}.$$

Let $\mathcal{F}^t$ be the constraint set of $\boldsymbol{P}$ at the $t$-th iteration of our algorithm. Note that $\mathcal{F} \subseteq \mathcal{F}^t$ by definition. If the algorithm stops at the $t$-th iteration, then for any $\{\boldsymbol{a}_i \mid \boldsymbol{a}_i \in [-1,1]^{AC}, i \in [k]\}$, $\boldsymbol{P}^t$ satisfies

$$g(\boldsymbol{P}^t; \boldsymbol{a}_1, \cdots, \boldsymbol{a}_k) \le 0,$$

which implies $\boldsymbol{P}^t \in \mathcal{F}$. Consequently,

$$f(\boldsymbol{P}^t) = \max_{\boldsymbol{P} \in \mathcal{F}^t} f(\boldsymbol{P}) \ge \max_{\boldsymbol{P} \in \mathcal{F}} f(\boldsymbol{P}) \ge f(\boldsymbol{P}^t).$$

As a result, $f(\boldsymbol{P}^t) = \max_{\boldsymbol{P} \in \mathcal{F}} f(\boldsymbol{P})$ so $\boldsymbol{P}^t$ is an optimal solution of $\mathsf{FairFront}_k(\alpha_{\text{SP}}, \alpha_{\text{EO}}, \alpha_{\text{OAE}})$.

If the algorithm never stops, consider any convergent sub-sequence of $\boldsymbol{P}^t$ that converges to a limit point $\boldsymbol{P}^* \in \mathcal{T}(C|AC)$. To simplify our notation, we assume $\boldsymbol{P}^t \to \boldsymbol{P}^*$ as $t \to \infty$. Since $\{\mathcal{F}^t\}_{t \geq 1}$ is non-increasing and they all contain $\mathcal{F}$, there exists a set $\mathcal{F}^*$ such that
$$\lim_{t \to \infty} \mathcal{F}^t = \mathcal{F}^*, \quad \mathcal{F} \subseteq \mathcal{F}^*.$$
Therefore, we have
$$f(\boldsymbol{P}^*) = \lim_{t \to \infty} f(\boldsymbol{P}^t) = \lim_{t \to \infty} \max_{\boldsymbol{P} \in \mathcal{F}^t} f(\boldsymbol{P}) = \max_{\boldsymbol{P} \in \mathcal{F}^*} f(\boldsymbol{P}).$$
Since $\mathcal{F} \subseteq \mathcal{F}^*$, we have
$$f(\boldsymbol{P}^*) = \max_{\boldsymbol{P} \in \mathcal{F}^*} f(\boldsymbol{P}) \geq \max_{\boldsymbol{P} \in \mathcal{F}} f(\boldsymbol{P}).$$
If $\boldsymbol{P}^* \notin \mathcal{F}$, then there exists a $(\bar{\boldsymbol{a}}_1, \cdots, \bar{\boldsymbol{a}}_k)$, such that $g(\boldsymbol{P}^*; \bar{\boldsymbol{a}}_1, \cdots, \bar{\boldsymbol{a}}_k) > 0$. Let $(\boldsymbol{a}_{1,t}, \cdots, \boldsymbol{a}_{k,t})$ be the output of Step 2 at $t$-th iteration. Since $\boldsymbol{P}^* \in \mathcal{F}^t$ for all $t$, we have
$$g(\boldsymbol{P}^*; \boldsymbol{a}_{1,t}, \cdots, \boldsymbol{a}_{k,t}) \leq 0. \tag{31}$$
By the optimality of $(\boldsymbol{a}_{1,t}, \cdots, \boldsymbol{a}_{k,t})$, we have
$$g(\boldsymbol{P}^t; \boldsymbol{a}_{1,t}, \cdots, \boldsymbol{a}_{k,t}) \geq g(\boldsymbol{P}^t; \bar{\boldsymbol{a}}_1, \cdots, \bar{\boldsymbol{a}}_k). \tag{32}$$
Suppose that some sub-sequence of $(\boldsymbol{a}_{1,t}, \cdots, \boldsymbol{a}_{k,t})$ converges to a vector $(\boldsymbol{a}_1^*, \cdots, \boldsymbol{a}_k^*)$. For the sake of simplicity, we assume $(\boldsymbol{a}_{1,t}, \cdots, \boldsymbol{a}_{k,t}) \to (\boldsymbol{a}_1^*, \cdots, \boldsymbol{a}_k^*)$ as $t \to \infty$. On the one hand, taking limit of $t \to \infty$ on both sides of (32) leads to
$$g(\boldsymbol{P}^*; \boldsymbol{a}_1^*, \cdots, \boldsymbol{a}_k^*) \geq g(\boldsymbol{P}^*; \bar{\boldsymbol{a}}_1, \cdots, \bar{\boldsymbol{a}}_k).$$
On the other hand, taking limit of $t \to \infty$ on both sides of (31) leads to
$$g(\boldsymbol{P}^*; \boldsymbol{a}_1^*, \cdots, \boldsymbol{a}_k^*) \leq 0.$$
Therefore,
$$0 \geq g(\boldsymbol{P}^*; \boldsymbol{a}_1^*, \cdots, \boldsymbol{a}_k^*) \geq g(\boldsymbol{P}^*; \bar{\boldsymbol{a}}_1, \cdots, \bar{\boldsymbol{a}}_k) > 0,$$
which is impossible. Therefore, $\boldsymbol{P}^* \in \mathcal{F}$ and, as a result, we have
$$f(\boldsymbol{P}^*) = \max_{\boldsymbol{P} \in \mathcal{F}^*} f(\boldsymbol{P}) \geq \max_{\boldsymbol{P} \in \mathcal{F}} f(\boldsymbol{P}) \geq f(\boldsymbol{P}^*) \implies \max_{\boldsymbol{P} \in \mathcal{F}} f(\boldsymbol{P}) = f(\boldsymbol{P}^*).$$
$\square$

### B.4   Additional Results

We establish basic properties of $\mathcal{C}$ and $\mathsf{FairFront}(\alpha_{\text{SP}}, \alpha_{\text{EO}}, \alpha_{\text{OAE}})$ in the following lemma.

**Lemma 3.** $\mathcal{C}$ *is a convex subset of* $\mathcal{T}(C|AC)$ *and* $\mathsf{FairFront}(\alpha_{\text{SP}}, \alpha_{\text{EO}}, \alpha_{\text{OAE}})$ *is a concave function w.r.t.* $\alpha_{\text{SP}}, \alpha_{\text{EO}}, \alpha_{\text{OAE}}$. *Here the constants* $A$ *and* $C$ *denote the number of protected groups and the number of classes.*

Next, we discuss a special case—X is discrete—under which $\mathcal{C}$ has a simple characterization.

**Remark 3.** If X is a *discrete* variable with a *finite support* $[D]$, we can write $P_{\text{X}|\text{S},\text{Y}}$ as a transition matrix $\boldsymbol{\Phi} \in \mathcal{T}(D|AC)$. By introducing an auxiliary variable $\boldsymbol{M} \in \mathcal{T}(C|D)$, we can write $\boldsymbol{P} \in \mathcal{C}$ equivalently as linear constraints: $\boldsymbol{P} = \boldsymbol{\Phi} \boldsymbol{M}$ by using the last condition of Lemma 1. Consequently, Proposition 1 boils down to a linear program. However, this characterization fails to generalize to continuous data because $\boldsymbol{\Phi}$ and $\boldsymbol{M}$ will have an infinite dimension; for categorical data, this characterization suffers from the curse of dimensionality since the support size of X grows exponentially fast w.r.t. the number of features.

## C   Details on the Experimental Results

### C.1   Additional Experiments

In this section, we present additional experimental results to further support our findings. We reproduce our experimental results on the German Credit dataset (Bache and Lichman, 2013) and HSLS (High School Longitudinal Study) dataset (Ingels et al., 2011; Jeong et al., 2022) in Figure 4 and Figure 5. In particular, the HSLS dataset experiment is a multi-group, multi-label experiment. Our observation is consistent with those on the previous two datasets—the fairness-accuracy curves given by SOTA fairness interventions, such as `Reduction` and `FairProjection`, are close to the information-theoretic limit.

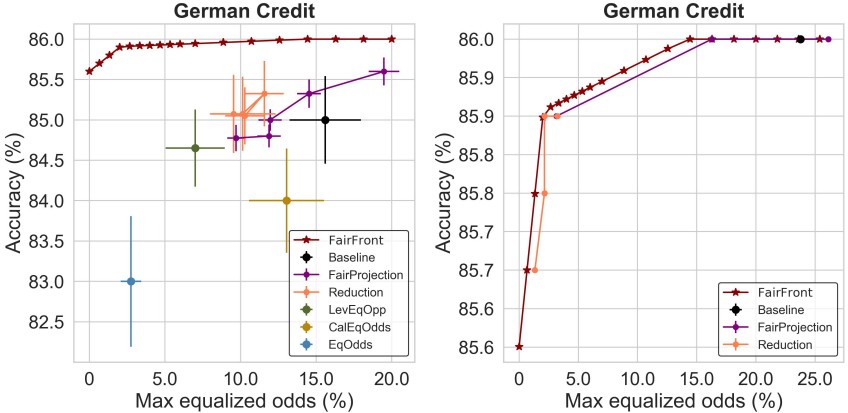

Figure 4: We reproduce our experiments on the German Credit dataset. Our observation is consistent with those on the previous two datasets—the fairness-accuracy curves given by SOTA fairness interventions, such as `Reduction` and `FairProjection`, are close to the information-theoretic limit.

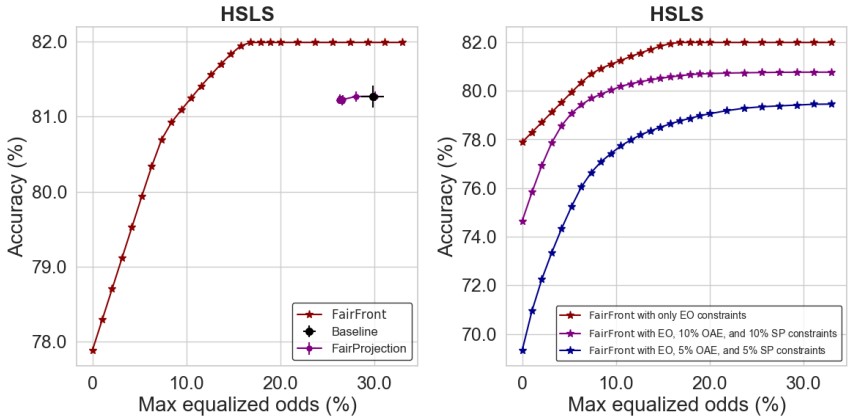

Figure 5: We reproduce our experiments on the HSLS dataset with multi-group and multi-label pre-processing. On the right, we also demonstrate that `FairFront` can take into account multiple fairness considerations at once. We show how the fairness-accuracy curve changes as we add new types of group fairness constraints (i.e., adding OAE and SP constraints in addition to EO).

## C.2 Dataset

**Adult.** We use sex (female or male) as the group attribute and income ($> 50K$ or $<= 50K$) as the target for prediction. We use sex, hours-per-week, education-num, age, marital status, relationship status (husband or wife) as the input features—we include the group attribute as an input feature. We group age into a total of 12 disjoint intervals: $[0, 20), [20, 25), \cdots, [65, 70), [70, \infty)$; we group hours-per-week into a total of 14 disjoint intervals: $[0, 10), [10, 15), \cdots, [65, 70), [70, \infty)$.

**COMPAS.** We use race (African-American or Caucasian) as the group attribute and is_recid (recid. or no recid.) as the target for prediction. We use race, age, c_charge_degree, sex, priors_count, c_jail_in, c_jail_out as the input features—we include the group attribute as an input feature. We use the last two features by taking their difference to be their length_of_stay. We remove entries where COMPAS case could not be found (is_recid = -1) and entries with inconsistent arrest information. We also binarize sex and remove traffic offenses. We quantize age the same way we do in the Adult dataset and quantize length_of_stay by every 30 days and let 0 be a separate category.

**German Credit.** We use age (below or above 25 years old) as the group attribute and the credit column, which represents whether the loan was a good decision, as the target for prediction. We use

loan duration in month, credit amount, age, number of existing credits at this bank, sex, credit history, savings, and length of present employment as input features. We include the group attribute age as an input feature. We group credit amount into three disjoint intervals: [0, 5000), [5000, 10000), [10000,∞). We group duration of loan into two categories: under 36 months and over 36 months.

**HSLS.** We use race as the group attribute and mathematics test score (number of questions answered correctly out of 72) as the target for prediction. This is a multi-group and multi-label dataset. The entire population is grouped by 4 categories: White, Asian, African American, and Others. We seek to predict the mathematics test performance from a set of attributes, including the scale of student's mathematical identity, scale of student's mathematics utility, scale of one's mathematics self-efficacy, parent's education, parent's income, scale of student's sense of school belonging, race, and sex. Note that we include the group attribute as an input feature. We group the target column (estimated number of questions answered correctly) into a total of 5 disjoint intervals: $[0, 30)$, $[30, 40), [40, 50), [50, 60), [60, \infty)$; we group the scale of student's mathematical identity, mathematics utility, mathematics self-efficacy, and sense of school belonging into a total of 4 disjoint intervals, characterized by standard deviations away from the mean: $(-\infty, -1), [-1, 0), [0, 1), [1, \infty)$.

### C.3 Benchmark

Each benchmark method's hyper-parameter values are provided below. Each point in Figure 2 for `Baseline`, `EqOdds`, `CalEqOdds`, `Reduction`, `LevEqOpp`, and `FairProjection` is obtained by applying the obtained classifier to 10 different test sets. For the Adult dataset, we use Random Forest with n_estimators=15, min_samples_leaf=3, criterion = log_loss, bootstrap = False as our baseline classifier; for the COMPAS dataset, we use Random Forest with n_estimators = 17 as our baseline classifier. For the German Credit dataset, we use Random Forest with n_estimators=100,min_samples_split =2,min_samples_leaf=1 as our baseline classifier. They are all implemented by Scikit-learn (Pedregosa et al., 2011).

**EqOdds (Hardt et al., 2016).** We use AIF360 implementation of `EqOddsPostprocessing` and the default hyper-parameter setup.

**CalEqOdds (Pleiss et al., 2017).** We use AIF360 implementation of `CalibratedEqOddsPostprocessing` and the default hyper-parameter setup.

**Reduction (Agarwal et al., 2018).** We use AIF360 implementation of `ExponentiatedGradientReduction`. We vary the allowed fairness constraint violation $\epsilon \in \{0.001, 0.01, 0.2, 0.5, 1, 2, 5, 10, 15\}$ for Adult dataset and $\epsilon \in \{0.001, 0.01, 0.2, 0.5, 1, 2, 5, 10, 15\}$ for Adult with missing values. We vary $\epsilon \in \{0.001, 2, 5, 10, 15, 20, 25, 30, 35, 40\}$ for COMPAS to obtain a fairness-accuracy curve, and $\epsilon \in \{0.001, 0.1, 0.5, 1, 2, 7, 8, 10, 15, 20, 25, 30\}$ for COMPAS with 50% missing values in the minority group. We use $\epsilon \in \{20, 50, 80, 95\}$ for German Credit dataset and $\epsilon \in \{5, 8, 10, 20, 23\}$ when using Bayes Optimal classifier.

**LevEqOpp (Chzhen et al., 2019).** We use the Python implementation of `LevEqopp` from the Github repo in Alghamdi et al. (2022). We follow the same hyperparameters setup as in the original method.

**FairProjection Alghamdi et al. (2022).** We use the implementation from the Github repo in Alghamdi et al. (2022) and set use_protected = True. We use Random Forest with n_estimators = 17 as the baseline classifier to predict S from $(X, Y)$. We set the list of fairness violation tolerance to be $\{0.07, 0.075, 0.08, 0.085, 0.09, 0.095, 0.1, 0.5, 0.75, 1.0\}$ for Adult dataset and $\{0.02, 0.03, 0.04, 0.05, 0.06, 0.07, 0.08, 0.1, 0.5, 1.0\}$ for COMPAS dataset to obtain a fairness-accuracy curve. We set the list of fairness violation tolerance to be $\{0.005, 0.01, 0.02, 0.07, 0.1, 0.15\}$ on the German Credit dataset experiment, and $\{0.0001, 0.001, 0.005, 0.01, 0.015, 0.02, 0.05\}$ when using a Bayes optimal baseline classifier. For the HSLS dataset, the list of tolerance is $\{0.06, 0.065, 0.07, 0.08, 0.1, 0.09, 0.15, 0.2, 0.3\}$.

| | Multiclass | Multigroup | Avoid disparate treatment | Multi-constraint | Curve |
|---|:---:|:---:|:---:|:---:|:---:|
| Hardt et al. (2016) | ✗ | ✓ | ✗ | ✗ | ✗ |
| Corbett-Davies et al. (2017) | ✗ | ✓ | ✗ | ✗ | ✗ |
| Menon and Williamson (2018) | ✗ | ✗ | ✓ | ✗ | ✓ |
| Chzhen et al. (2019) | ✗ | ✗ | ✓ | ✗ | ✗ |
| Yang et al. (2020) | ✓ | ✓ | ✗ | ✓ | ✓ |
| Zeng et al. (2022a) | ✗ | ✓ | ✗ | ✗ | ✓ |
| Zeng et al. (2022b) | ✗ | ✓ | ✗ | ✗ | ✗ |
| Our approach | ✓ | ✓ | ✓ | ✓ | ✓ |

Table 2: Comparison with existing work that investigate the fairness Pareto frontier. **Multiclass/multigroup**: can handle multiclass classification problems with multiple protected groups; **Avoid disparate treatment**: not require the classifier to use group attributes as an input variable; **Multi-constraint**: can handle multiple (group) fairness constraints simultaneously; **Curve**: produce fairness-accuracy trade-off curves (instead of a single point).

# D   More on Related Work

We provide a detailed comparison with existing work on fairness Pareto frontier and ML uncertainty in this section.

## D.1   Fairness Pareto Frontier

We present in Table 2 a detailed comparison of our approach with previous studies that have investigated the fairness Pareto frontier and fair Bayes optimal classifier. In short, our approach is different from this line of research as it simultaneously combines several important aspects: it is applicable to *multiclass* classification problems with *multiple* protected groups; it *avoids disparate treatment* by not requiring the classifier to use group attributes as an input variable; and it can handle *multiple* fairness constraints simultaneously and produce fairness-accuracy trade-off curves (instead of a single point).

## D.2   Aleatoric and Epistemic Uncertainty

In this paper, we divide algorithmic discrimination into aleatoric and epistemic discrimination. We borrow this notion from ML uncertainty literature (see Hüllermeier and Waegeman, 2021, for a survey). Here we provide a detailed comparison between them.

In terms of their definitions, epistemic uncertainty arises from a lack of knowledge about the best model, such as the Bayes predictor, while epistemic discrimination results from a lack of knowledge about the optimal "fair" predictive model. On the other hand, aleatoric uncertainty is the irreducible part of uncertainty caused by the random relationship between input features and label, while aleatoric discrimination is due to inherent biases in the data-generating distribution.

In terms of their characterization, epistemic uncertainty can in principle be reduced by including additional information (e.g., more data); epistemic discrimination can be reduced in a similar approach since a data scientist can choose a more effective fairness-intervention algorithm with access to more information.

Finally, in the infinite sample regime, a consistent learner will be able to remove all epistemic uncertainty, assuming the model class is large enough and there are no computational constraints. Analogously, we demonstrate in Figure 1 that when the underlying distribution is known, SOTA fairness interventions are able to eliminate epistemic discrimination as their fairness-accuracy curves are close to the fair front.

# E   More on Future Work

In this paper, we present an upper bound estimate for FairFront in Algorithm 1. It is important to note that this estimate may be subjected to errors originating from various sources. These include (i)

the approximation error of the function $g$, (ii) estimation errors from computing the expectation in (7) with a finite dataset, and (iii) the influence of hyperparameters, $T$ (number of running iterations of Algorithm 1) and $k$ (number of segments in the piece-wise linear functions). Regarding the dependence on $T$, our Theorem 2 ensures the algorithm's asymptotic convergence as $T \to \infty$. However, we have not established a proof for its behavior at a finite $T$. Regarding the dependence on $k$, we conjecture that $k = A * C$ should suffice, where $A$ is the number of protected groups and $C$ is the number of labels. While Blackwell proved this result for $k = 2$ in Theorem 10 of Blackwell (1953), an extension of this proof to a general value of $k$ appears to remain an open problem.

We define aleatoric and epistemic discrimination with respect to the entire population. Investigating their per-instance counterparts and the relationship to individual fairness would be a compelling area of future research. Additionally, a more nuanced analysis of aleatoric and epistemic discrimination is desirable, further breaking them down into fine-grained components. For instance, epistemic discrimination may be attributed to various factors including limited training data, noisy observations of labels or sensitive attributes, and limitations of learning algorithms. Finally, investigating other criteria, such as scalability, generalization, and robustness in evaluating existing fairness interventions is a significant topic for future exploration.

