# OpenReview forum: "Aleatoric and Epistemic Discrimination: Fundamental Limits of Fairness Interventions"
_NeurIPS.cc/2023/Conference — NeurIPS 2023 spotlight_

### Official Review · Reviewer_D24A · 2023-06-26

**Soundness:** 3 good
**Presentation:** 2 fair
**Contribution:** 2 fair
**Rating:** 4
**Confidence:** 5

**Summary:**

The manuscript introduces FairFront i.e., an estimation for the upper bound on the Pareto Frontier for Fairness and Accuracy. The authors empirically show the tightness of this bound by showing how SOTA approaches perform close to the FairFront, while the gap may be attributed to the distributional variations.

**Strengths:**

- The authors tackle the problem of finding the upper bound on the Pareto frontier in contrast to prior works which focus on the lower bound.
- The approach presented in the manuscript is theoretically grounded (by the essence of using Blackwell's results).
- SOTA approaches are used to show the tightness of the bound empirically.

**Weaknesses:**

- The biggest weakness is that epistemic and aleatoric discrimination are terms introduced in the paper which have a link to the corresponding concepts in the uncertainty literature. However, the way these terms are used contradicts the definitions of the uncertainty counterparts e.g., Aleatoric uncertainty is linked to missing data (line 14-15, 65-66, 250-251). In prior literature, the lack of data falls in the domain of epistemic uncertainty [1][2][3].
- In the manuscript, Aleatoric discrimination is linked to distributional differences. However, prior work either differentiates it from both aleatoric and epistemic uncertainty[4] or considers it as a part of epistemic uncertainty [5 (Section 9.2 dedicated to this topic)].
- It is mentioned that distributional differences are largely a grey area in prior work. However, works such as FairBatch [6] and FairMixup [7], tackle such cases and are ideal for inclusion in the experiments.
- The manuscript is a bit hard to follow. Even though it has content for describing which components are used formally, there is little intuitive background. Some links are hard to follow. e.g. it is not immediately obvious that $(S, Y) - X - \hat{Y})$ refers to the Markov chain in Definition 2. It is not evident why this is used as the Markov chain.


[1] Swiler, Laura P., Thomas L. Paez, and Randall L. Mayes. "Epistemic uncertainty quantification tutorial." _Proceedings of the 27th International Modal Analysis Conference_. 2009.
[2] Shaker, Mohammad Hossein, and Eyke Hüllermeier. "Aleatoric and epistemic uncertainty with random forests." _Advances in Intelligent Data Analysis XVIII: 18th International Symposium on Intelligent Data Analysis, IDA 2020, Konstanz, Germany, April 27–29, 2020, Proceedings 18_. Springer International Publishing, 2020.
[3] https://docs.aws.amazon.com/prescriptive-guidance/latest/ml-quantifying-uncertainty/epistemic-uncertainty.html
[4] Amini, Alexander, et al. "Deep evidential regression." _Advances in Neural Information Processing Systems_ 33 (2020): 14927-14937.
[5] Varshney, Kush R. "Trustworthy Machine Learning." _Chappaqua, NY_ (2021).
[6] Roh, Yuji, et al. "FairBatch: Batch Selection for Model Fairness." _International Conference on Learning Representations_.
[7] Mroueh, Youssef. "Fair Mixup: Fairness via Interpolation." _International Conference on Learning Representations_. 2021.

**Questions:**

It is hard to see intuitively why FairFront works e.g., why is the markov chain $(S, Y) - X- \hat(Y)$? How does it filter out valid classifiers?

I acknowledge that I read the response and I feel overall the rebuttal was satisfying.

**Limitations:**

No.

- From Lemma 1, it appears that one of the limitations of this work is that the FairFront is only applicable to convex classifiers.
- I would highly suggest renaming the terms 'epistemic' and 'aleatoric' discrimination since they do not portray the corresponding uncertainty counterparts accurately and may lead to confusion. Terms such as 'Distributional Discrimination' may be more fitting.

---

> ### Author Rebuttal · Authors · 2023-08-09
>
> We appreciate the reviewer’s their careful reading of our paper and thoughtful comments!
>
> ---
> **Q1. Epistemic and aleatoric discrimination and their link to uncertainty literature.**
>
> A1. We thank the reviewer for highlighting this crucial point. Below, we discuss their link to uncertainty literature.
>
> In terms of their definitions, epistemic uncertainty arises from a lack of knowledge about the best model, such as the Bayes predictor, while epistemic discrimination results from a lack of knowledge about the optimal fair predictive model. On the other hand, aleatoric uncertainty is the irreducible part of uncertainty caused by the random relationship between input features and label, while aleatoric discrimination is also the irreducible part due to inherent biases in the data-generating distribution.
>
> In terms of their characterization, epistemic uncertainty can in principle be reduced by including additional information; epistemic discrimination can be reduced in a similar approach by e.g., adding more data, as a data scientist can choose a more effective fairness-intervention algorithm with access to more information.
>
> In the infinite sample regime, a consistent learner will be able to remove all epistemic uncertainty, assuming the model class is large enough and there are no computational constraints. Analogously, we demonstrate in Fig. 5 that when the underlying distribution is known, SOTA fairness interventions are able to eliminate epistemic discrimination as their fairness-accuracy curves are close to the fair front.
>
> You are right: the lack of data falls in the domain of epistemic uncertainty. However, the missing data we are discussing is not about missing rows, but rather missing feature values. Indeed, this aligns with the literature on uncertainty, see Fig. 5 on page 466 of [Hüllermeier and Waegeman, 2020] for a discussion on how missing feature values can amplify aleatoric uncertainty.
>
> References:
>
> --Hüllermeier, E. and Waegeman, W., 2021. Aleatoric and epistemic uncertainty in machine learning: An introduction to concepts and methods.
>
> ---
> **Q2. Aleatoric discrimination and distributional differences.**
>
> A2. We’d like to clarify that distributional shift is a form of epistemic discrimination. To understand this, recall that aleatoric discrimination is only based on the properties of the deployed data distribution. It is quantified under the assumption of having complete knowledge of this data distribution (that is, as if an infinite amount of deployed data is available). In contrast, distributional shift arises due to the imperfect knowledge of the deployed data distribution as we can only observe $P_{train}$ which differs from $P_{deploy}$. Hence, by observing more samples from $P_{deploy}$, $P_{train}$ would be “closer” to $P_{deploy}$, and as a result, fairness interventions trained on $P_{train}$ and tested on $P_{deploy}$ could demonstrate an improved fairness-accuracy curve. This would lead to a reduction in epistemic discrimination.
>
>
>
>
> ---
> **Q3. Distributional differences are largely a grey area in prior work. However, works such as FairBatch and FairMixup, tackle such cases and are ideal for inclusion in the experiments.**
>
> A3. We thank the reviewer for pointing out the missing references, which will be cited in the revised paper. As mentioned in A2, distributional difference is related to the form of epistemic discrimination. Hence, these two works align with the line of fairness interventions aimed at reducing epistemic discrimination, and we will discuss them in the updated paper.
>
> ---
> **Q4. Little intuitive background. Not obvious that $(S,Y) – X – \hat{Y}$ refers to the Markov chain in Def. 2.**
>
> A4. Thank you for highlighting the issue and we will provide more background information in the revised paper. Regarding the Markov chain, it is a rather simple condition and holds if $\hat{Y}$ is generated from $X$ (that is, the classifier only uses $X$ as input). This is always the case in practice – otherwise the classifier would be rather trivial (i.e., it would use $Y$ to predict itself). Also, observe that $S$ can be incorporated as a feature of $X$, so there is no loss of generality in this assumption.
>
> ---
> **Q5. Hard to see intuitively why FairFront works e.g., why is the markov chain $(S,Y) – X – \hat{Y}$? How does it filter out valid classifiers?**
>
> A5. Here is a short summary about how $FairFront$ works:
>
> First, computing $FairFront$ directly is intractable as it requires optimizing over a large, or even infinite-dimensional function space. To circumvent this issue, we write $FairFront$ as a function of a transition matrix $P_{\hat{Y}|S,Y}$ and use $\mathcal{C}$ to denote the set of all $P_{\hat{Y}|S,Y}$ that correspond to a feasible classifier. This is a much lower dimensional problem since it does not directly depend on the cardinality of the input features $X$. The markov chain $(S,Y) – X – \hat{Y}$ eliminates the transition matrices that are not associated with a feasible classifier (please refer to Remark 2 in Appendix B.4 on page 17 for an illustrative example). By leveraging Blackwell’s results, we approximate the convex set $\mathcal{C}$ via piecewise linear functions with coefficients $a_i$. Algorithm 1 outlines a method that alternatively tightens the approximation of $\mathcal{C}$ and computes $FairFront$ under this approximation.
>
> ---
> **Q6. From Lem. 1, it appears that the FairFront is only applicable to convex classifiers.**
>
> A6. We would like to clarify that FairFront characterizes the fundamental fairness-accuracy trade-offs among ALL types of classifiers, not just the convex ones. The function $\phi$ in Lemma 1 is not a classifier but rather will be approximated by piecewise linear functions in Algorithm 1 for characterizing $\mathcal{C}$ in Definition 2.
>
> ---
> **Q7. Epistemic and aleatoric discrimination and the corresponding uncertainty counterparts.**
>
> A7. Please refer to our response to your Q1 and Q2.

---

> > ### Comment · Reviewer_D24A · 2023-08-15
> >
> > A1:
> > Thanks for the clarification. The missing features makes sense in the case of aleatoric uncertainty. Can a link be defined in terms of this uncertainty and Fairfront mathematically? Or is the distribution used as a surrogate to link them?
> >
> > A2:
> > Since you agree that distribution shift is a form of epistemic discrimination, how would aleatoric discrimination be linked to Fairfront in light of this? As per the rebuttal of A1, I am having trouble drawing a formal link between Fairfront and aleatoric uncertainty.

---

> > > ### Author Response · Authors · 2023-08-15
> > >
> > > Thank you so much for your response!
> > >
> > > A1. Below, we revisit the mathematical definition of aleatoric uncertainty according to Section 2.2 on page 463 of [Hüllermeier and Waegeman, 2020]. We also recall the definition of FairFront and contrast the two based on their definitions.
> > >
> > > ---Given the probability distribution of the deployed data $P_{X,Y}$ and a loss function $\ell$, the Bayes optimal classifier is defined as
> > > $$
> > > f^* := \arg\min_f E_{P_{X,Y}}[\ell(f(X)), Y]
> > > $$
> > > where the minimization is over all measurable functions $f:\mathcal{X} \to \mathcal{Y}$. Then aleatoric uncertainty is defined as the uncertainty of applying $f^*$ to predict the outcome of a new test point $x_{test}$.
> > >
> > >
> > > ---The Bayes optimal *fair* classifier is a naturally extension of $f^*$ with discrimination control taken into account:
> > > $$
> > > f_{fair}^* := \arg\min_{f} E_{P_{X,Y}}[\ell(f(X)), Y]\ s.t. \text{DiscVio}(f) \leq \alpha
> > > $$
> > > where $\text{DiscVio}(f)$ measures the discrimination violation of the classifier $f$ (see Table 1 for some examples). The value of $f_{fair}^*$ will inherently depend on the fairness level $\alpha$, leading to a Pareto Frontier of fairness and accuracy, and this frontier is exactly FairFront.
> > >
> > > Both aleatoric uncertainty and FairFront, by definition, assume full knowledge about the data distribution $P_{X,Y}$. Hence, they are considered irreducible and cannot be diminished by collecting more data. However, incorporating more features into the model can enhance the performance of the Bayes optimal classifier, since you are inherently changing the data distribution by changing $X$. Similarly, if $X$ is observed through a noisy process (e.g., entries of $X$ are erased), then reducing the noise (e.g., number of erasures) – which would again change $P_{X,Y}$ – would affect aleatoric uncertainty. Similarly, it also impacts the fairness-accuracy curve of the Bayes optimal *fair* classifier. Hence, including more features in the model can help reduce both aleatoric discrimination (delineated by FairFront) and aleatoric uncertainty.
> > >
> > > Regarding missing values, consider the following example directly inspired by the HSLS dataset used in our paper. Data is collected from anonymous student questionnaire answers ($X$) in order to predict student performance and school dropout risk ($Y$). Each questionnaire is assumed to be independent and drawn from the same distribution $P_{X,Y}$.  If there are a limited number of students to query, the limited sample size translates to imperfect knowledge of $P_{X,Y}$, and hence epistemic uncertainty/discrimination. This uncertainty could be reduced by querying more students, thus increasing sample size and rendering a more precise estimate of $P_{X,Y}$ and of the best prediction accuracy for a given fairness level.
> > >
> > > Now, assume that certain students may be reluctant to answer some questions in the questionnaire. For example, students whose parents did not go to college may leave the section on parents' education completely blank. In other words, for each questionnaire, some questions may be left blank, i.e. $X$ may include erased features. This cannot be overcome by querying more students, since missing features are built into the data generating distribution $P_{X,Y}$. In this case, missing values consist of *aleatoric* uncertainty. Of course, we could probe a student and ask them to complete the questionnaire but, since in our example answers are anonymous, we do not have the option. Collecting data from additional students would not resolve this issue, and the missing features are part of the aleatoric uncertainty/discrimination.
> > >
> > > In summary, it is important to distinguish the sources of missing data in terms of limited samples vs missing features:
> > >
> > >
> > > --- Lack of data due to *limited sample:* $P_{X,Y}$ is not known exactly, leading to epistemic uncertainty/discrimination. In this case, collecting more samples from $P_{X,Y}$ reduces epistemic uncertainty/discrimination since it leads to a more precise estimate of the distribution.
> > >
> > > –- Lack of data due to *missing features:* the features $X$ can be missing. In this case, since the data generating distribution $P_{X,Y}$ itself may yield missing features, this uncertainty cannot be reduced by drawing more samples from $P_{X,Y}$. Here, missing features are part of aleatoric uncertainty and discrimination.
> > >
> > > **(continued in the next comment ...)**

---

> > > > ### Author Response · Authors · 2023-08-15
> > > >
> > > > A2. In response to your question, we provide the following example to link distribution shift with aleatoric discrimination.
> > > >
> > > > Consider a bank that enters a new market and finds that its credit score system underperforms for customers aged over 60. This discrepancy may be due to the credit score model being trained on a dataset from a market with a different distribution, perhaps influenced by local income levels. To mitigate this bias, the bank could gather more data from the new market and apply post-processing fairness interventions to its credit scoring system. Since collecting more data from the new market improves accuracy and fairness, the distribution shift posed by the new market consists of an epistemic source of discrimination.
> > > >
> > > > However, collecting new data would not address aleatoric discrimination. Aleatoric discrimination presumes full knowledge about the data distribution in the deployment environment.  For example, say that in the new market customers in group A always pay-off their loans, whereas customers in group B always default. In this case, achieving statistical parity in credit scoring across both groups would require compromising overall accuracy. This is due to the distribution of the data in the new market, and simply adding more samples from the new market wouldn't resolve this trade-off. Hence, it consists of aleatoric discrimination.
> > > >
> > > > We will add the above discussions to the paper and are happy to answer any follow-up questions.

---

> > > > > ### Comment · Reviewer_D24A · 2023-08-16
> > > > >
> > > > > The idea of finding an upper bound on the pareto frontier is novel and new. However, the way aleatoric and epistemic uncertainty are divided needs more attention and the link appears to be handwavy. For aleatoric uncertainty, while it is true, missing features can be a cause, there are also other factors, such as latent noise in the measurement. The overall idea is great but a bit more is needed to draw the connection between uncertainty types and Fairfront.
> > > > >
> > > > > Adding more details: Since the upper bound is theoretical, the uncertainties have to also be formally linked to them. The danger of using examples to link theoretical concepts is that they do not cover all use cases e.g. missing features are not the only source of aleatoric uncertainty. Is the epistemic uncertainty counterpart of the bound really epistemic and is the aleatoric one really aleatoric? The handwavy part is the missing solid formal link which would ideally be a parametric equation consisting of FairFront into its aleatoric and epistemic counterparts. How are the two disentangled? That is also an open problem which leads to more questions. Is the disentanglement problem also solved by this work?
> > > > >
> > > > >
> > > > > Final Verdict: borderline reject

---

> > > > > > ### Author Response · Authors · 2023-08-20
> > > > > >
> > > > > > **Disentangle aleatoric and epistemic discrimination via their mathematical definition**
> > > > > >
> > > > > > Thank you for your comments. We provided illustrative examples in our previous response to clarify the difference between aleatoric and epistemic discrimination – we misunderstood your point of being qualitative. We reiterate below the precise mathematical definition of aleatoric and epistemic discrimination to differentiate these two concepts.
> > > > > >
> > > > > > [**Aleatoric discrimination**].  For a given data distribution $P_{X,Y}$, (group) fairness metrics of choice (e.g., statistical parity and equalized odds), and discrimination violation threshold $\alpha \geq 0$, we quantify aleatoric discrimination by $1 – FairFront(P_{X,Y}, \alpha)$, which is
> > > > > > $$
> > > > > > 1 - \max_{f} E_{P_{X,Y}}[\mathbb{I}(f(X) = Y)]\ s.t. \text{DiscVio}(f) \leq \alpha
> > > > > > $$
> > > > > > Here the maximization is over all measurable functions $f:\mathcal{X} \to \mathcal{Y}$. In other words, aleatoric discrimination quantifies the minimal achievable loss under discrimination control over all possible classifiers. This only depends on the underlying distribution $P_{X,Y}$. As mentioned above, this aligns with the definition of aleatoric uncertainty in [Hüllermeier and Waegeman, 2020].
> > > > > >
> > > > > > By its definition, aleatoric discrimination is irreducible by adding more data (as it already assumes perfect knowledge of the data distribution $P_{X,Y}$) or altering model class and learning algorithm (as it optimizes over all measurable functions). It is an inherent property of the distribution $P_{X,Y}$.
> > > > > >
> > > > > > [**Epistemic discrimination**]. For a fairness-intervention algorithm $\mathcal{A}$ that takes a training set $\mathcal{D} = \set{(x_i, y_i), i=1,...,n}$ as input and outputs a classifier $h$ from a hypothesis class $\mathcal{H}$, we quantify epistemic discrimination by comparing FairFront with the discrimination violation and accuracy of $h$:
> > > > > > $$
> > > > > > FairFront(P_{X,Y}, \alpha) – Acc(h)
> > > > > > $$
> > > > > > where $\alpha = \text{DiscVio}(h)$. In other words, epistemic discrimination measures the difference between the (accuracy, discrimination control) attained by the classifier from a fairness intervention and the optimal achievable one, delineated by FairFront.
> > > > > >
> > > > > > Note that training and deployed data may be drawn from different probability distributions (i.e., the generating distribution of $(x_i, y_i)$ is different from $P_{X,Y}$). Hence, distribution shift is a form of epistemic discrimination. Some other sources of epistemic discrimination include but are not limited to the choice of fairness-intervention algorithm, number of data, and the choice of model class. In general, epistemic discrimination arises from a lack of knowledge about the optimal “fair” model and can be eliminated in the limit of infinite data from the true (test) distribution $P_{X,Y}$ (see Figure 5 in Appendix).
> > > > > >
> > > > > > We will add this to the revised paper.
> > > > > >
> > > > > > -----
> > > > > >
> > > > > > **Different sources of aleatoric discrimination**
> > > > > >
> > > > > > We can apply the aforementioned mathematical definitions to investigate various sources of aleatoric discrimination. In this context, we introduce a new theorem that sheds light on some of these sources.
> > > > > >
> > > > > > Theorem. Suppose that $X_{obs} = g(X)$ where $g$ can be any (potentially randomized) function. Then $FairFront(P_{X_{obs}, Y}, \alpha) \leq FairFront(P_{X,Y},\alpha)$ for all $\alpha \geq 0$.
> > > > > >
> > > > > > This theorem can be proved by the data processing inequality. This theorem shows that aleatoric discrimination can arise if the observed variable $X_{obs}$ is a noisy observation of $X$. The function $g$ in the above theorem can be an erasure channel that mask feature values according to a certainty probability (i.e., missing features or unobserved predictive variables) or introduce noise to the input features (i.e., noisy measurements as you mentioned).
> > > > > >
> > > > > > Above we only provide two sources of aleatoric discrimination. In general, aleatoric discrimination is influenced by the properties of the data distribution, so any action that might alter this distribution can have an impact on aleatoric discrimination.

---

### Official Review · Reviewer_7W2X · 2023-06-28

**Soundness:** 4 excellent
**Presentation:** 3 good
**Contribution:** 2 fair
**Rating:** 6
**Confidence:** 3

**Summary:**

This paper splits discrimination in machine learning into aleatoric (which is that inherent to the data distribution), and epistemic (which is that due to choices in the model). They use Blackwell’s results to characterize the fairness Pareto frontier curve. Then, on 4 datasets with 5 fairness interventions, they characterize how close to their upper bound the algorithms are able to achieve.

**Strengths:**

- Comprehensive set of experiments on a number of datasets and fairness interventions
- Clear writing and presentation of work
- Claims are backed up by theories and proofs

**Weaknesses:**

- Given that the output from the algorithm (L236) is always the upper bound for FairFront, rather than exactly FairFront, I wish the paper would have been more upfront with this throughout, especially in the introduction and abstract, because I found this a bit misleading
- Even though the names of aleatoric and epistemic discrimination are taken from prior work, I think they should be potentially rethought because for example, “epistemic discrimination” feels like a term that should be restricted to the philosophical foundations of the word “epistemology,” e.g., as it is used in epistemic injustice (https://academic.oup.com/book/32817), rather than a not-so related mathematical constraint
- Given that a primary differentiation from prior work (L76) is that it works on multiclass classification as well as multiple protected groups, I would have liked to see empirical results on this rather than just claiming this is true
- Given how similar the motivation of this work is to Chen, I. et al. (NeurIPS 2018)’s, I would have liked to see a deeper comparison of the findings of both works
- “Discrimination” as used in this paper is only restricted to different forms of measurement disparities, e.g., statistical parity, equalized odds, and excludes many other forms of algorithmic discrimination that are more structural. So, sentences like L35 “We divide algorithmic discrimination into two categories” should be caveated that this is only a very specific form of algorithmic discrimination that is being divided into two categories (https://aclanthology.org/2020.acl-main.485/)
  - As further explanation, taxonomies of algorithmic discrimination are its own genre of normative research, for example the distinction between representational vs allocational harms (https://www.youtube.com/watch?v=fMym_BKWQzk), or a taxonomy of harms within representational harms (https://arxiv.org/abs/2305.01776)

**Questions:**

Listed above in weaknesses.

**Limitations:**

I think that the authors could be more upfront about the limitation that their work maps out the upper bound of the FairFront rather than the FairFront exactly, and also that they have taken a rather narrow lens of fairness, neglecting broader societal concerns.

---

> ### Author Rebuttal · Authors · 2023-08-09
>
>
> We thank the reviewer for the kind comments and the encouragement!
>
> ---
> **Q1. Algorithm (L236) is always the upper bound for FairFront. More upfront with this throughout.**
>
> A1. Thank you for raising this concern. We will clarify that we provide an upper bound estimate of $FairFront$ in the introduction and abstract.
>
> ---
> **Q2. The names of aleatoric and epistemic discrimination.**
>
> A2. We appreciate your insightful comment regarding the potential ambiguity associated with our terminology. We will address this confusion in the revised paper. For clarity, we borrow these terms from the uncertainty literature (see Appendix D.2). Therein, epistemic uncertainty refers to the reducible uncertainty stemming from a lack of knowledge about the best model, while aleatoric uncertainty is the inherent, irreducible uncertainty due to random associations between input features and labels. Analogously, epistemic discrimination arises from a lack of knowledge about the optimal fair predictive model, whereas aleatoric discrimination pertains to the irreducible aspects attributed to inherent biases in the data-generating distribution.
>
> ---
> **Q3. Empirical results on multiclass classification and multiple protected groups.**
>
> A3. Please refer to Figure 4 in Appendix where we conducted experiments on the HSLS dataset with 4 groups and 5 labels. In short, our observation is consistent: state-of-the-art fairness interventions are effective at reducing epistemic discrimination as their fairness-accuracy curves are close to FairFront.
>
> ---
> **Q4. Deeper comparison with Chen et al. (NeurIPS 2018).**
>
> A4. Chen et al. (2018) decomposed group fairness measures into bias, variance, and noise in their Theorem 1 and proposed strategies for reducing each term accordingly. There are two notable differences compared with their work.
>
> First, Theorem 1 in Chen et al. (2018) provides a decomposition for only a *single* group fairness metric. In contrast, our FairFront provides a more comprehensive analysis, characterizing the fundamental trade-offs between *multiple* group fairness metrics and *accuracy* among all classifiers. This analysis is more technically complex, as the interactions between different group fairness metrics and accuracy can incur tradeoffs, or in some cases, may even be mutually exclusive (as evidenced by the impossibility results [Kleinberg etal., 2016; Chouldechova, 2017]).
>
> Second, the applications of our method are different from Chen et al. (2018). We specifically applied our method to benchmark existing fairness interventions, showcasing their effectiveness in eliminating epistemic discrimination across group fairness metrics. The analysis in Chen et al. (2018) is not suited for this purpose, as it solely considers a singular fairness metric. Additionally, we studied how the presence of missing values can impact aleatoric discrimination, thereby diminishing the effectiveness of fairness interventions. Note that the challenge posed by missing values has been generally neglected in the existing literature, including in the work of Chen et al. (2018). We hope that our efforts can inspire further research into this subject, contributing to the development of algorithms aimed at reducing biases under these conditions.
>
> ---
> **Q5. “Discrimination” used in this paper is only restricted to different forms of measurement disparities.**
>
> A5. We appreciate the reviewer for highlighting this crucial issue. We will explicitly state that our focus is limited to a specific form of algorithmic discrimination, namely, performance disparity across protected groups; other types of algorithmic discrimination or situations where group attributes are not clearly defined would require a non-trivial extension of our results.
>
> ---
> **Q6. Taxonomies of algorithmic discrimination are its own genre of normative research, for example the distinction between representational vs allocational harms, or a taxonomy of harms within representational harms.**
>
> A6. We thank the reviewer for pointing out these important references. As mentioned above, we will highlight that our focus is on statistical aspects of a specific form of algorithmic discrimination (performance disparities). We will also add the references you mentioned, specifically on the distinction between representation and allocational harms, as well as on active research on disentangling the range of normative concerns often bundled as "unfairness" or "discrimination." We will also highlight that, as posed by Katzman et al. (2023), no single measurement approach for "fairness" is definitive, particularly across different contexts and use cases.
>
> ---
> **Q7. More upfront about the limitation that their work maps out the upper bound of the FairFront. A rather narrow lens of fairness, neglecting broader societal concerns.**
>
> A7. Thank you for raising these important points. We hope our response in A1, A5, and A6 has addressed all of your comments. Please feel free to let us know if you have any additional comments that can help us further improve the paper.

---

> > ### Comment · Reviewer_7W2X · 2023-08-12
> >
> > Thank you to the authors for responding to my comments. Upon reading the other reviews, I hope that the authors will be very explicit about the limitations of what kind of “fairness” they cover in their work, as you have stated in your rebuttal. I appreciate your detailed response for the comparison to Chen et al. (2018), and hope that some of it can be in the main paper as well. I will keep my rating as it originally is.

---

> > > ### Author Response · Authors · 2023-08-12
> > > **Thank you for your prompt response!**
> > >
> > > Thank you for your prompt response! Yes, we will make sure to explicitly state what kind of "fairness" measures covered in this paper and include a detailed discussion with Chen et al., 2018 in the revised paper. Finally, we would like to express our appreciation once more for the insightful and constructive comments you provided.

---

### Official Review · Reviewer_wKi3 · 2023-07-04

**Soundness:** 3 good
**Presentation:** 3 good
**Contribution:** 4 excellent
**Rating:** 7
**Confidence:** 3

**Summary:**

This paper proposes a decomposition of discrimination (in ML classifiers) into aleatoric (irreducible) and epistemic (reducible) components. The paper surveys related work in fairness. It then introduces and discusses the Fairness Pareto Frontier, which is essentially an upper bound on the accuracy of the best possible fairness constrained classifier. The paper proposes an algorithm for estimating aleatoric discrimination (this upper bound) which utilizes Blackwell’s results on comparing statistical experiments. It then demonstrates the applicability of the algorithm through experiments on 5 datasets. Finally, it identifies missing values in the dataset as a source of aleatoric discrimination,  providing experimental evidence to support the discussion.

**Strengths:**

Though I was admittedly not familiar with Blackwell’s results, the paper’s technical contributions are insightful and convincing.
The paper makes a novel connection to statistical results that (as far as I am aware) had not been considered previously in fairness research.
The theoretical framework and results, as well as the estimation method are applicable to a wide range of fairness-constrained predictive modelling problems (multi-class, multi-attribute), and thus are a substantial contribution to the field.
The paper does a good job of surveying related work, and distinguishing its contributions.
The writing is concise and of high quality.


**Weaknesses:**

The paper is dense and would benefit considerably from an illustrative example, helping to build intuition about the terms and concepts introduced. Specifically to help digest the main theoretical result, Theorem 1, and the piece-wise linear approximation.

The paper presents FairFront as an upper bound whose estimate depends on k, and T. However, for a given data distribution, it is unclear how the k-dependence is affected by the quantity of available data. As a practitioner, I would want to understand the robustness of this upper bound to data availability. Some experiments that subsample the dataset would be a helpful starting point. Perhaps a version of the experiment discussed in Appendix C.2 with varying levels of data availability (up to the infinite data regime).

The region described as epistemic discrimination (if Figure 1) is effectively a gap between an upper bound (FairFront) and a lower bound (some fairness-constrained modelling approaches). As such, the true fairness Pareto Frontier is somewhere in between. It is unclear why the reader should interpret FairFront as the more accurate bound, and thus interpret the gap as epistemic discrimination (rather than FairFront estimation error). That is to say, the paper could elaborate on the tightness of this approximate upper bound based on choices of k and T.

The use of transparency in Figure 2 makes it difficult to compare scenarios (missing probabilities), especially when printed.


**Questions:**

When the fairness constraints are trivially satisfied, for example $\alpha_{\text{SP}}, \alpha_{\text{EO}}, \alpha_{\text{OAE}} = \inf$, how should one interpret the FairFront?

Line 231: what is meant by “mostly violated”?


**Limitations:**

The paper refers to fairness and discrimination, but makes no effort to connect these concepts to the harms or benefits experienced by real people. This is fairly common practice in the field and not unique to this paper. However, it is nonetheless a noteworthy limitation. That is to say, one cannot truly comment on the fairness (or discrimination) of a machine learning system without a broader understanding of the social impacts–beyond just its predictions.

It is unclear whether the method can be applied in the numerous important situations whereby the prediction outcome is unobserved (or partially observed), and what effects estimating these counterfactuals would have on the method, e.g., in lending–where the outcomes of the rejected cohort are counterfactuals.

---

> ### Author Rebuttal · Authors · 2023-08-09
>
> We thank the reviewer for the thoughtful review and for appreciating the merits of the work!
>
> ---
> **Q1. An illustrative example, helping to build intuition about the terms and concepts introduced.**
>
> A1. We thank the reviewer for this valuable suggestion. One concrete example is the COMPAS dataset. It has been overused in the field, leading to recent calls to move on from it as a benchmark. Nevertheless, since its public release, hundreds of group fairness interventions have been proposed and benchmarked on this dataset, amounting to hundreds of achievable fairness-accuracy pairs. A natural question to ask is: are we close to the optimal Pareto Frontier on this dataset? FairFront – a theoretical upper bound for the *best* achievable fairness accuracy frontier – proves that we are indeed close to the optimal frontier when these datasets reflect the true underlying distribution. This provides theoretical backing to the sentiment of moving on from such datasets and optimizing solely for group fairness and accuracy.
>
> Please also refer to Appendix B.4 for two technical examples that illustrate our results in special cases (Remark 2: $X$ and $(S,Y)$ are independent; Remark 3: $X$ is discrete).
>
> ---
> **Q2. The estimate of FairFront depends on $k$ and $T$. It is unclear how the k-dependence is affected by the quantity of available data.**
>
> A2. This is a great suggestion! Regarding k-dependence, we conjecture that $k = A * C$ should suffice, where $A$ is the number of protected groups and $C$ is the number of labels. While Blackwell proved this result for $k=2$ in his 1953 paper's Theorem 10, he didn't extend his proof for a general $k$. In the context of infinite data, our experiments have numerically verified this conjecture, and it has consistently held true. As for T-dependence, we observed in our experiments that setting $T = 20$ always ensured algorithm convergence.
>
> ---
> **Q3. The gap between an upper bound (FairFront) and a lower bound (fairness-constrained approaches). The tightness of this approximate upper bound based on choices of $k$ and $T$.**
>
> A3. You are right: the gap can be the result of the estimation error or epistemic discrimination. Nonetheless, we observed in our experiments that the estimation error tends to be quite small when $k\geq A*C$ and $T=20$. Broadly, the theoretical characterization of the estimation error of FairFront, as well as its dependence on $k$ and $T$ remains an open question. We will address this issue in the limitations section of our revised paper.
>
> ---
> **Q4. The use of transparency in Figure 2.**
>
> A4. We apologize for this inconvenience and will use distinct colors to represent different missing probabilities.
>
> ---
> **Q5. When $\alpha_{SP}, … = \inf$, how to interpret FairFront?**
>
> A5. When the fairness constraints are trivially satisfied, FairFront evaluates the accuracy of the Bayes optimal classifier.
>
> ---
> **Q6. What is meant by “mostly violated”?**
>
> A6. Suppose $P$ does not belong to $\mathcal{C}_k$. We will find a piecewise linear function that partitions the space into two regions. One region contains $P$ and the other one contains $\mathcal{C}_k$. “Mostly violated” means we construct this function by *maximizing* the distance between $P$ and the boundary defined by the function.
>
> ---
> **Q7. The paper refers to fairness and discrimination but makes no effort to connect these concepts to the harms or benefits experienced by real people.**
>
> A7. Thank you for highlighting this critical aspect. We fully agree: understanding the real-world impacts and harms on real people goes well beyond the limited technical and mathematical metrics often emphasized in ML research.
>
> Our paper's main message is that existing fairness interventions optimized solely for group fairness and validated on overused datasets like Adult and COMPAS are provably nearing their theoretical best in terms of specific group fairness metrics and accuracy. Rather than just adding to such interventions, our results provide theoretical backing to your sentiment that it's time for the community to move on from incremental improvements on these benchmarks. For example, as we argue in Section 4.2, addressing real-world challenges, like data with missing values where these patterns correlate with group attributes, is crucial. Our findings partially close the chapter on the myopic approach of designing interventions optimized entirely for group fairness metrics and benchmarking on overused datasets: in this limited setting – for which hundreds of interventions have been produced – we are already close to the information-theoretic best!
>
> You're right; we could have delved deeper into the broader social implications of our findings. In our final remarks, we'll stress the importance of moving beyond just numbers to consider the tangible effects on real individuals and the practical factors impacting data quality – this was indeed the core motivation for our work, and we hope to send a signal that we should focus on other problems beyond simply group fairness and accuracy on datasets such as COMAPS. We value your feedback and will emphasize this perspective in the revised manuscript.
>
> ---
> **Q8. Whether the method can be applied when the prediction outcome is unobserved (or partially observed).**
>
> A8. Thank you for pointing out this scenario. Indeed, our technique can be applied to estimate FairFront where prediction outcomes are unobserved or partially observed. This is because Algorithm 1 only requires a classifier $g$ that predicts $(S,Y)$ from $X$, along with the probability distribution of unlabeled data for computing the expectation. If the prediction outcome is unobserved, one could learn the function $g$ using a one-class classification method. However, we caveat that the approximation error of $g$ may increase, potentially affecting the accuracy of estimating FairFront.

---

> > ### Comment · Reviewer_wKi3 · 2023-08-15
> >
> > I appreciate the detailed response. I have read the other reviews and associated rebuttals. I stand by my score.

---

> > > ### Author Response · Authors · 2023-08-16
> > >
> > > Thank you so much for your response. We will make sure to include the promised changes in the revision (both in the main text and appendix).

---

### Official Review · Reviewer_CXu9 · 2023-07-29

**Soundness:** 3 good
**Presentation:** 1 poor
**Contribution:** 2 fair
**Rating:** 4
**Confidence:** 2

**Summary:**

The paper casts the issue of fairness in machine learning from the perspective of two classes of discrimination: those due to aleatoric uncertainty, or where the inherent limitations of the data distribution, and epistemic discrimination, which is due to modeling choices. It then uses this framework to analyze a common set of fairness metrics, showing that epistemic discrimination can be reduced by optimizing for these fairness measures, but not necessarily aleatoric discrimination.

**Strengths:**

- The choice to disentangle sources of discrimination due to modeling choices vs. data properties is sound and reasonable.
- To the best of my knowledge, studying aleatoric discrimination via Blackwell's equations is novel (the reviewer is not familiar with this set of techniques, so can offer little here)
- The choice to study missing data values is natural.

**Weaknesses:**

My main concern with this paper was that its contribution feels quite limited both in terms of thoroughness of the experiments (it would have been interesting to study different sources of data properties beyond miss values, such as Khani & Liang, 2020) do with spurious features, as well as its relevance in modern machine learning (the only data studied is the well-known COMPAS dataset, and it could have been interesting to consider situations where more advanced models -- such as language or vision models -- are applied, as well as settings where the motivation to apply ML is well justified unlike risk assessment).

I also think the authors could clarify quite early on they specific a particular subset of fairness interventions - group fairness with defined subgroup labels. This is not always possible when subgroups aren't clearly defined, challenging to measure, or are intractable.

In a similar vein, the writing could be more precise to properly convey the scope of FairFront. For example, sentences like " For instance, if the training set contains few samples from a given population group, then increasing sample diversity is a more effective strategy than  selecting a more complex model class or training strategy." --> there are training methods that try to account for unequal group size (e.g. Hashimoto, 2018)) when group labels are inaccessible, and increasing sample diversity is hard. These sentences could just be qualified more that we are in a particular setting where we know the groups we want to be fair over, and can measure them.

**Questions:**

Are you able to run experiments that involve a bit more complex sources of data variability (e.g. a natural language task with different dialects, so more than 2 group labels, and using more state-of-the-art models like GPT series)?

**Limitations:**

I think the paper could also be more clear on the importance of the overall result. Lacking in thorough experimentation, the paper largely presents a framework but does not argue for its utility. Do the authors intend their framework to be used as a rigorous benchmark? If so, they should focus on more comprehensive set of experiments. Otherwise, what is the core message of knowing that methods that are not data-specific do not address dataset-specific sources of discrimination?

---

> ### Author Rebuttal · Authors · 2023-08-09
>
> We thank the reviewer for the kind comments and the encouragement!
>
> ---
> **Q1. Contribution feels limited both in terms of experiments and its relevance in modern machine learning.**
>
> A1. Thank you for raising this important point. First, please note that we provide numerical results for **four** benchmark datasets: Adult, COMPAS, German Credit, and HSLS. Both German Credit and HSLS results are presented in Appendix C. We highlight that the HSLS dataset is a multi-class classification task with multiple protected groups that first appeared in the ML literature last year, and captures a common use-case of ML in education (student performance prediction, see Jeong etal., 2022). We also compared FairFront against **five** group fairness interventions (see 265-278). These include Reduction and FairProjection which are, to our knowledge, the state-of-the-art in terms of fairness-accuracy frontiers. The scale of our numerical results is comparable to recent comprehensive benchmarks of fairness interventions with publicly available reproducible code (cf. Alghamdi etal. in NeurIPS 2022).
>
> We highlight that our choice of presenting Adult and COMPAS in the main text was strategic. These datasets, and especially COMPAS, have been extensively used as benchmarks in fairness research. Our findings on them show that there are *diminishing returns* in benchmarking new fairness interventions on these datasets and that existing methods approach the information-theoretically optimal Pareto frontier given by FairFront (see line 323). We believe this sends a pivotal, theoretically-grounded message to the community to innovate beyond these overused datasets. The choice of these two datasets for the main text was also practical: most interventions can be readily applied to them without change, allowing for more comprehensive benchmarks of existing methods against FairFront (see the five methods in Figure 1).
>
> We certainly appreciate the suggestion on exploring more advanced models and other data properties. It aligns well with our vision, as we touch upon in lines 26-27 and in our final remarks. Our intent is to drive the field towards addressing fresh challenges in responsible ML rather than refining on established datasets. Again, a main contribution of our paper is that there is no room for improvement in terms of vanilla group fairness/accuracy values of these overused datasets. As a field, we should perhaps close this research chapter and move to new challenges in responsible ML such as the ones you mentioned.
>
> ---
> **Q2. Group fairness with defined subgroup labels. This is not always possible when subgroups aren't clearly defined, challenging to measure, or are intractable.**
>
> A2. Thank you for pointing this out – we will clarify this in the introduction. As we mentioned above, one of our main goals is to demonstrate that the subset of fairness interventions optimized solely for group fairness metrics where groups are well-defined appear to achieve the information-theoretic best on standard benchmarks. There are hundreds of such interventions proposed in the past decade, and we hope our results provide theoretical backing for the field to move on to more realistic and pressing issues such as the ones you suggest. The case where subgroups aren't clearly defined (or represented by a set of functions, such as in multiaccuracy and multicalibration) is not covered by our results, but is indeed an important research direction. We will stress this limitation in the final section.
>
> ---
> **Q3. There are training methods that try to account for unequal group size (e.g. Hashimoto, 2018)) when group labels are inaccessible, and increasing sample diversity is hard.**
>
> A3. You are correct, and we will qualify the scope of FairFront in a revised manuscript. Specifically, we will add that:
>
> "When group labels are inaccessible or only partially accessible, increasing sample diversity can be challenging. In such cases, fairness interventions that account for only partially observed group attributes (e.g., Hasimoto, 2018) are a compelling alternative."
>
> ---
> **Q4. Run experiments that involve more complex sources of data variability (e.g. a natural language task with different dialects, so more than 2 group labels, and using more state-of-the-art models like GPT series)?**
>
> A4. Thank you for your suggestion about exploring tasks like natural language with varied dialects and using advanced models such as the GPT series. We note that our scope is tabular datasets, since this allows us to benchmark state-of-the-art fairness interventions against the theoretical optimum given by FairFront.
>
> While gathering dialect-based datasets and training and/or probing LLMs is challenging given our limited resources, we reiterate the additional results in Appendix C that include other datasets (e.g., HSLS, where there are multiple groups). As we mentioned in the response to your Q1, a central theme in our paper is to demonstrate that existing fairness interventions are approaching the information-theoretic optimal fairness-accuracy Pareto frontier on widely-used datasets. We are excited to include your suggestion in our concluding section as a compelling direction for future work, particularly in light of our findings on the diminishing returns of focusing on overused tabular datasets.
>
> ---
> **Q5. Importance of the overall result. Lacking in thorough experimentation. What is the core message?**
>
> A5. Please refer to our responses to your Q1, Q2, and Q4. In short, the core message in this study is that there are *diminishing returns* in benchmarking new fairness interventions on standard (overused) datasets as existing methods are approaching the information-theoretically optimal Pareto frontier delineated by FairFront. We back this claim with numerical results from 4 benchmark datasets, tested against 5 state-of-the-art fairness interventions.

---

> > ### Comment · Reviewer_CXu9 · 2023-08-18
> > **Response**
> >
> > Thank you for your response.
> >
> > Q1. I appreciate the clarification regarding the strategic choice in choosing commonly used datasets (e.g. Adult) to highlight the limitation of  studying these datasets given the FairFront bound. I don't think this messaging comes across in the current paper draft, however -- the focus of the paper seems more of a critique of broader approaches in the community to fairness interventions, rather than a specific critique of these simple datasets. If anything, I would thinking experiments with a more complex dataset and highlighting differences in results would drive home this point more. So I would encourage the authors to think more about the framing.
> >
> > Q3. The purpose of this comment was more to highlight that there exist proposed methods in the literature that are *training* time interventions, not just dataset-level operations -- the current text seems to imply addressing unequal dataset issues is impossible via the training method, but modifies loss functions can capture this.
> >
> > I have updated my score, but would strongly encourage the authors to explicitly state their choice to use their datasets in a revised version.

---

> > > ### Author Response · Authors · 2023-08-18
> > >
> > > Thank you so much for your response and follow-up comments!
> > >
> > > Q1. Absolutely. In the revised paper, we will clearly state our main message: there are diminishing returns in benchmarking new fairness interventions on standard (overused) datasets as existing methods are approaching the information-theoretically optimal Pareto frontier delineated by FairFront. Additionally, we will articulate the rationale behind selecting commonly used datasets for our experiments.
> > >
> > > Q3. Thank you again for highlighting this line of work. We will clarify our scope more clearly in the revised paper.

---

### Official Review · Reviewer_iE9m · 2023-08-01

**Soundness:** 4 excellent
**Presentation:** 4 excellent
**Contribution:** 4 excellent
**Rating:** 8
**Confidence:** 4

**Summary:**

In this manuscript, the authors make two main contributions to the technical study of algorithmic fairness.

Firstly, on the conceptual level, they propose to distinguish between aleatoric and epistemic discrimination. By the former term, they refer to the notion that the optimal achievable performance level may differ between different (e.g., racial or gender-based) groups, corresponding to notions of differing task difficulty. No algorithmic bias mitigation approach can resolve aleatoric discrimination, which is a property of a dataset, not a model. Epistemic discrimination, on the other hand, refers to a model that performs sub-optimally on a given group, compared to the performance level it could achieve on this dataset.

Secondly, the authors develop a new way to characterize an upper bound on the Fairness Pareto frontier, i.e., the frontier characterizing optimal trade-offs between overall model accuracy and various fairness constraints. The characterization is based on an old theoretical result by Blackwell and facilitates an efficient implementation by means of a greedy algorithm developed by the authors. This second contribution relates to the first in that it provides a lower bound on the level of aleatoric discrimination in a given dataset.

Several numerical experiments in standard (low-dimensional, tabular) algorithmic fairness datasets confirm the validity of the bound and the relative tightness of some previously proposed bias mitigation techniques, emphasizing that significantly better trade-offs are unlikely to be achievable in these datasets. An experiment with artificially induced missing data shows how this increases aleatoric discrimination, as would be expected.

**Strengths:**

The paper makes several highly original and important contributions to the algorithmic fairness field.

The framing of different sources of algorithmic discrimination as separated into aleatoric and epistemic discrimination is novel, and it importantly helps in understanding and characterizing the limitations of bias mitigation techniques, which can only ever aim to reduce epistemic discrimination. While similar distinctions have been made before, I find the framing and formulation presented in the manuscript particularly clear and useful.

To my knowledge, the manuscript provides the first characterization of an *upper* bound on the Pareto frontier, the latter being an essential object of study in algorithmic fairness. It also treats the general case of both multiple sensitive groups and multiple output labels, which is relatively rare in the field (which mostly focuses on the binary setting for both.)

The manuscript introduces a novel theoretical tool, based on the old results of Blackwell, into the field of algorithmic fairness, which may spark further theoretical innovations beyond the present paper. I consider this alone to be an important creative contribution.

The paper is very well written, the results are clearly presented, and prior work is comprehensively discussed and acknowledged.

**Weaknesses:**

I do not see many weaknesses in the manuscript, but there is one issue that I think deserves a more comprehensive discussion. This is related to the approximation of g, which maps inputs X to P(S,Y | X), where Y is the output label and S the sensitive group. As I understand it, the correctness of the approximated upper bound depends crucially on the correctness of this approximation. In the simple low-dimensional tabular data cases considered in the manuscript, g can be computed efficiently and precisely, but in the case of more complex (continuous, image?) domains, I would imagine this to be a serious limitation. Do I understand correctly that if g is approximated poorly, the computed bound may in fact not be an upper bound? I would appreciate a more comprehensive discussion of the impact of this approximation on the correctness of the estimated upper bound.

Also, can the authors provide some hints at important future extensions of the presented work, or important limitations of the approach? Section 5 is called "Final Remarks and Limitations", but it actually only summarizes the contributions of the present manuscript and lists no limitations.

**Questions:**

1. As outlined in detail above: what are the implications of the approximation on g? How does this impact potential other application domains, such as image analysis? Do I understand correctly that in the tabular cases considered in the paper, g is not approximated but in fact computed exactly?
2. I was a little confused by the notation in some places. For instance, the authors write that they "define FairFront(αSP, αEO, αOAE) as the solution of the following optimization problem", where the following optimization problem optimizes over a classifier h. However, later on, the authors use notation such as "FairFront1(αSP, αEO, αOAE) ≥ FairFront2(αSP, αEO, αOAE)", prompting me to ask: what exactly *is* "FairFront"? Is it the model? Its achieved accuracy? Something else? This could be made clearer by a notation such as "FairFront = arg max ...", for instance.
3. How does the complexity of the algorithm scale with, e.g., the number of sensitive groups to be considered? The authors emphasize that their approach "can be easily extended to the setting where multiple subgroups overlap"; how exactly would this work? In the case of multiple non-binary sensitive attributes, the number of subgroups to consider grows combinatorially; will that present practical problems?
4. To give practitioners some sense of the complexity of Algorithm 1, could the authors provide run time measurements for the case studies?

**Limitations:**

1. Like outlined above, the limitations induced by approximating g are presently not obvious to me; I could imagine these to be quite consequential if they effectively limit the applicability of the presented work to low-dimensional tabular data.
2. More generally, the authors do not discuss any limitations of their approach. One limitation that comes to my mind is that it has been shown repeatedly that fairness-accuracy trade-offs may be illusory in the case of group-dependent label noise or selection biases, see, e.g., Blum and Stangl (2020), Wick et al. (2019), Dutta et al. (2020). Another limitation seems to be that the presented approach relies on the availability, correctness, and validity of sensitive attributes, which is often not given; see, e.g., Jacobs and Wallach (2021) and Tomasev et al. (2021).

Blum and Stangl (FORC 2020): Recovering from Biased Data: Can Fairness Constraints Improve Accuracy? https://drops.dagstuhl.de/opus/volltexte/2020/12019/pdf/LIPIcs-FORC-2020-3.pdf

Dutta et al. (ICML 2020): Is There a Trade-Off Between Fairness and Accuracy? A Perspective Using Mismatched Hypothesis Testing. https://proceedings.mlr.press/v119/dutta20a.html

Jacobs and Wallach (FAccT 2021): Measurement and Fairness. https://dl.acm.org/doi/10.1145/3442188.3445901

Tomasev et al. (FAccT 2021): Fairness for Unobserved Characteristics: Insights from Technological Impacts on Queer Communities. https://dl.acm.org/doi/10.1145/3461702.3462540

Wick et al. (NeurIPS 2019): Unlocking Fairness: a Trade-off Revisited. Unlocking Fairness: a Trade-off Revisitedhttps://papers.nips.cc/paper/2019/file/373e4c5d8edfa8b74fd4b6791d0cf6dc-Paper.pdf

---

> ### Author Rebuttal · Authors · 2023-08-09
>
> We thank the reviewer for the thoughtful comments and for appreciating the novelty of the work!
>
> ---
> **Q1. Approximation of $g$ and the impact on the estimated upper bound.**
>
> A1. Yes, your understanding is correct! The approximation error of $g$ can influence the estimation of $FairFront$. If $g$ is not approximated accurately, the estimation might not serve as an upper bound any longer.
>
> To circumvent this issue, we applied fairness interventions to the entire dataset in our experiments and subsequently resampled 30% of the data for the test set (see lines 276-278). In this case, $g$ can be precisely computed from the empirical distribution without any error, and FairFront gives an information-theoretic upper bound. Generally, for low-dimensional tabular data, we anticipate that the approximation error of $g$ won't be significant, given the relative ease of training well-calibrated base models like random forests to predict S and Y from X. We acknowledge that in other domains, this approximation error might be significant. A practical solution would be bootstrapping confidence intervals for refitting the curve over data splits. Nonetheless, a theoretical characterization of the extent to which this error impacts the estimation of $FairFront$ remains an open problem (please refer to our A2 for discussions about the limitations which will be included in the last section).
>
> ---
> **Q2. Future extensions and important limitations.**
>
> A2. Thank you for highlighting this matter. In response to your concerns, we provide a discussion on the limitations and potential future directions below. We will ensure this discussion is incorporated into the revised paper.
>
> In this paper, we present an upper bound estimate for $FairFront$. However, it is important to note that this estimate may be subjected to errors originating from various sources. These include (i) the approximation error of the function $g$, (ii) estimation errors from computing the expectation in Eq. (6) with a finite dataset, and (iii) the influence of hyperparameters, $T$ (number of running iterations of Algorithm 1) and $k$ (number of segments in the piecewise linear functions). Regarding the dependence on $T$, our Theorem 2 ensures the algorithm's asymptotic convergence as $T \to \infty$. However, we have not established a proof for its behavior at a finite $T$. Regarding the dependence on $k$, we conjecture that $k = A * C$ should suffice, where $A$ is the number of subgroups and $C$ is the number of labels. While Blackwell proved this result for $k=2$ in his 1953 paper's Theorem 10, an extension of this proof to a general value of $k$ is an open problem.
>
> We define aleatoric and epistemic discrimination with respect to the entire population. Investigating their per-instance counterparts and the relationship to individual fairness would be a compelling area of future inquiry. Additionally, a more nuanced analysis of aleatoric and epistemic discrimination is desirable, further breaking them down into fine-grained components. For instance, epistemic discrimination may be attributed to various factors including limited training data, noisy observations of labels or sensitive attributes, and constraints of learning algorithms. Characterizing each of these components and devising appropriate solutions to mitigate them can lead to a more comprehensive taxonomy of sources of bias and (un)fairness in classification and prediction. Lastly, further exploration of other evaluation criteria, such as scalability, generalization, and robustness against partial knowledge of group attributes in the context of benchmarking existing fairness interventions is a valuable avenue for future research.
>
> ---
> **Q3. Implications of the approximation on $g$ and how this impacts potential other application domains.**
>
> A3. Please refer to A1.
>
> ---
> **Q4. What exactly is FairFront?**
>
> A4. Thank you for bringing up this concern. To clarify, $FairFront(\alpha)$ measures the maximal *achieved accuracy* of the model, given that its discrimination violation is upper bounded by $\alpha$.
>
> Mathematically,
> $$FairFront(\alpha_{SP}, …) := \max_{h} Accuracy(h)\ s.t. SP(h) \leq \alpha_{SP}, ...$$
>
> $FairFront_{k}$ is an upper bound approximation of $FairFront$ using $k$-piecewise linear functions. We will clarify this in the revised paper.
>
> ---
> **Q5. Scalability with the number of sensitive groups.**
>
> A5. As the number of protected groups, denoted by $A$, increases, both the number of variables in the convex program and the DC program in Alg. 1 increase linearly with $A$. Note that the running time of standard convex/DC program solvers is only mildly dependent on the number of variables (see e.g., the dimension-indep. convergence bounds in Cor. 4.1 in Abbaszadehpeivasti et al. 2023 and Thm. 5 in Faust et al. 2023). In our experiments, we also observed that even when doubling the number of groups and labels, 20 iterations consistently suffice for our algorithms to converge.
>
> –Abbaszadehpeivasti etal., 2023. On the rate of convergence of the difference-of-convex algorithm (DCA).
>
> –Faust etal., 2023. A Bregman Divergence View on the Difference-of-Convex Algorithm.
>
> ---
> **Q6. Provide run time measurements for the case studies**
>
> A6. Below, we present the runtime of Alg. 1 across various datasets of different scales. All experiments were run on a personal computer with 10 CPU cores and 16GB memory. We did not optimize our Python implementation (e.g., using GPUs) so the run time could be further reduced.
>
> German credit (1000 rows, 21 features): 0.53 mins
>
> COMPAS (5278 rows, 7 features): 1.73 mins
>
> Adult (46447 rows, 8 features): 6.11 mins
>
> HSLS (10937 rows, 9 features): 12.33 mins
>
> ---
> **Q7. The limitations induced by approximating g**
>
> A7. Please refer to A1.
>
> ---
> **Q8. Not discuss any limitations of their approach.**
>
> A8. Thank you for sharing your thoughtful insights and providing the associated references. Please refer to A2 for a detailed discussion regarding limitations.

---

> > ### Comment · Reviewer_iE9m · 2023-08-16
> >
> > I would like to thank the authors for their detailed and informative responses to my questions. In light of these, and also considering the other reviews, rebuttals, and discussions, I am updating my score to Strong Accept.
> >
> > One final remark concerning additional related work: it could also be interesting to mention the recent branch of the literature discussing omnipredictors. E.g., Globus-Harris et al. (2022) and Hu et al. (2023) discuss how and under which conditions Bayes-optimal fair classifiers can be derived using simple post-processing techniques from multicalibrated regressors.
> > https://arxiv.org/pdf/2209.07312.pdf
> > https://proceedings.mlr.press/v202/hu23b.html

---

> > > ### Author Response · Authors · 2023-08-16
> > >
> > > Thank you so much for your response and providing further related literature – indeed, this line of research is very relevant! We will expand the first paragraph of our Related Work section to highlight the work on omnipredictors and multicalibration, including the ones you mentioned, and their connection with Bayes-optimal fair classifiers. In our final section, we will also point to the burgeoning literature on omnipredictors and multicalibration as a source of new strategies for producing fair classifiers with theoretically-backed performance guarantees.

---

### Decision · Program_Chairs · 2023-09-21

**Decision:**

Accept (spotlight)

**Comment:**

This paper draws on the decomposition of uncertainty unto aleatoric and epistemic, to make a similar decomposition for algorithmic discrimination, and derive bounds on the Pareto frontier to provide lower bounds for the aleatoric discrimination. This relies on classical mathematical results by Blackwell, whose introduction to algorithmic fairness is novel and could lead to further new develowments.

The paper has received the reviewer scores 8, 4, 7, 6, 4, and the reviewers generally emphasize the same strengths in the paper. Of the accepts, two are quite enthusiastic about the paper, in particular highlighting its originality. Of the rejects, one would have liked to see more elaborate experiments on more complicated data, whereas the other finds that the use of the words "aleatoric" and "epistemic" differ from the way they are used in uncertainty modelling.

As the paper's main contributions are in the modelling and theoretical results, the lack of experiments on image or large language models is less of an an issue. While the authors are encouraged to consider whether wording could be in closer analogy with uncertainty modelling, this is not detrimental to what is otherwise a very strong paper.

The authors should include their promised updates to the final paper, and consider the reviews carefully and use them to further improve the clarity and accessibility of the paper.